# Adaptive spectroscopic visible-light optical coherence tomography for clinical retinal oximetry

Ian Rubinoff[1], Roman V. Kuranov[1,2], Raymond Fang[1], Zeinab Ghassabi[3], Yuanbo Wang[4], Lisa Beckmann[1], David A. Miller[1], Gadi Wollstein[3], Hiroshi Ishikawa[3,4], Joel S. Schuman [3] & Hao F. Zhang [1✉]

## Abstract

**Background** Retinal oxygen saturation ($sO_2$) provides essential information about the eye's response to pathological changes that can result in vision loss. Visible-light optical coherence tomography (vis-OCT) is a noninvasive tool that has the potential to measure retinal $sO_2$ in a clinical setting. However, its reliability is currently limited by unwanted signals referred to as spectral contaminants (SCs), and a comprehensive strategy to isolate true oxygen-dependent signals from SCs in vis-OCT is lacking.

**Methods** We develop an adaptive spectroscopic vis-OCT (ADS-vis-OCT) technique that can adaptively remove SCs and accurately measure $sO_2$ under the unique conditions of each vessel. We also validate the accuracy of ADS-vis-OCT using ex vivo blood phantoms and assess its repeatability in the retina of healthy volunteers.

**Results** In ex vivo blood phantoms, ADS-vis-OCT agrees with a blood gas machine with only a 1% bias in samples with $sO_2$ ranging from 0% to 100%. In the human retina, the root mean squared error between $sO_2$ values in major arteries measured by ADS-vis-OCT and a pulse oximeter is 2.1% across 18 research participants. Additionally, the standard deviations of repeated ADS-vis-OCT measurements of $sO_2$ values in smaller arteries and veins are 2.5% and 2.3%, respectively. Non-adaptive methods do not achieve comparable repeatabilities from healthy volunteers.

**Conclusions** ADS-vis-OCT effectively removes SCs from human images, yielding accurate and repeatable $sO_2$ measurements in retinal arteries and veins with varying diameters. This work could have important implications for the clinical use of vis-OCT to manage eye diseases.

## Plain language summary

Numerous diseases that cause blindness are associated with disrupted oxygen consumption in the retina, the part of the eye that senses light. This highlights the importance of accurately measuring oxygen consumption in the clinic. To address this challenge, we developed a method to analyze images of the retina which have been collected using visible-light optical coherence tomography, a non-invasive imaging method. Our approach achieves accurate oxygen level measurements in blood samples and in healthy volunteers. With further testing, our approach may prove useful in the clinical management of several diseases that cause blindness, allowing clinicians to more accurately diagnose disease and monitor the health of the eye.

[1] Department of Biomedical Engineering, Northwestern University, Evanston, IL 60208, USA. [2] Opticent Inc., Evanston, IL 60201, USA. [3] Department of Ophthalmology, New York University, New York, NY 10017, USA. [4] Currently with Department of Ophthalmology, Oregon Health & Science University, Portland, OR 97239, USA. ✉email: hfzhang@northwestern.edu

isual processing is one of the most oxygen-demanding functions in the human body[1,2]. Diseases such as glaucoma and diabetic retinopathy cause pathological changes in the eye, leading to irreversible vision loss and even blindness[2,3]. The retina regulates oxygen supply and extraction in response to pathological changes to satisfy altered metabolic demands[2–8]. Hence, changes in oxygen saturation (sO2) have the potential to be a sensitive biomarker for several retinal diseases and may be evident before irreversible vision loss occurs[3,9].

Optical coherence tomography (OCT) enabled noninvasive retinal imaging at a spatial resolution of a few micrometers[10–12] and has been considered as the "gold standard" for examining structural damages or therapeutic efficacy in nearly all vision-threatening diseases. However, low optical contrast in blood within the near-infrared (NIR) spectral range confounded OCT's sO2 measurements[13–15]. Recently, visible-light OCT (vis-OCT)[5,16–21] showed promise in overcoming the contrast limit because visible light is more sensitive to the optical absorption signatures of blood[13,22] than NIR light. Researchers obtained blood's depth-resolved, oxygen-dependent spectrum using a short-time Fourier transform (STFT)[16] of vis-OCT data. Such depth-resolved analysis by vis-OCT has the potential to isolate blood signals from other tissues, offering three-dimensional (3D) oxygenation measurements and improved accuracy, as compared with non-depth-resolved modalities such as fundus photography[23].

However, existing vis-OCT oximetry[16,21,24] does not take full advantage of depth information and remains limited in accuracy and repeatability, hampering its clinical impact. One fundamental limitation is the presence of unwanted signals referred to as spectral contaminants (SCs). We define SCs as any erroneous spectra not associated with blood's optical attenuation and classify them into sample-dependent and system-dependent SCs.

Figure 1 illustrates the human retina and sources of sample-dependent SCs. The spectral signatures of vis-OCT signals contain contributions from three groups of detected photons:

specular-reflected photons (group 1), backscattered photons not interacting with blood (group 2), and backscattered photons interacting with blood (group 3). Various photon interactions with tissues above or in a vessel, such as inner limiting membrane (ILM), retinal nerve fiber layer, vessel wall, and red blood cell (RBC) body, can add SCs to sO2 measurements.

System-dependent SCs come from the optical illumination, detection, and processing of the vis-OCT signals. We identified three key system-dependent SCs: spectrally-dependent roll-off (SDR), spectrally-dependent background bias (SDBG), and longitudinal chromatic aberration (LCA). Recently, we showed how SDR[25] and SDBG[26] contaminate spectroscopic measurements in imaging ex vivo blood samples and in vivo human retinas. LCA (illustrated by the green, yellow, and red colors in the scanning beam in Fig. 1) has also been shown as a contaminant for structural and spectroscopic OCT[27–29].

Previously, other groups and we analyzed vis-OCT backscattered signals from a vessel's posterior wall (PW) to measure sO2 in rodents[5,16,20,21] without fully correcting the abovementioned SCs. The fundamental limitation of such measurement is that it does not directly measure the attenuation coefficient of blood and is, therefore, susceptible to SCs. Furthermore, these methods cannot measure sO2 from vessels where the PW is undetectable due to strong blood attenuation in vessels with large diameters.

SCs are increasingly magnified and unpredictable in human imaging compared to small animal imaging due to physical and experimental constraints. In particular, human imaging must contend with reduced illumination powers for ocular laser safety and patient comfort, strong motions from the awake eyes and bodies, and larger variation in retinal anatomies[17,18,24,30]. Ideally, clinical retinal oximetry should be free from SCs and reliable across various vessel diameters.

In this work, we developed adaptive spectroscopic vis-OCT, or ADS-vis-OCT, which isolates blood's oxygen-dependent spectrum by conforming measurements to the unique properties of each vessel. We validated ADS-vis-OCT's accuracy in ex vivo samples made from bovine blood at 17 sO2 levels (Fig. S2). We then measured sO2 in 125 unique retinal vessels from 18 human volunteers imaged in a clinical environment. ADS-vis-OCT produced highly repeatable sO2 measurements (≤2.5%) across a broad range of vessel types and agreed within 2.1% of pulse oximeter readings in major arteries. Our sO2 results required no pre- or post-calibration to account for any systemic bias, suggesting that ADS-vis-OCT is insensitive to SCs or imaging environment. We compared ADS-vis-OCT to other reported vis-OCT sO2 techniques and found notable improvements, including a greater than threefold increase in measurement repeatability.

## Methods

**Principle of ADS-vis-OCT.** Figure 2 shows the flowchart for ADS-vis-OCT, where figures before and after each step depict their respective inputs and outputs. In brief, we used the STFTs to split high-resolution full-band vis-OCT image into 21 images at different spectral bands with reduced depth resolution in Step 1. Those narrow-band images were used to extract spectral signatures of the blood. In Step 2, we corrected the SBDG and co-register spectral B-scans. In Steps 3–7, we adaptively (unique for each vessel) identified the vis-OCT signal decay depth range to calculate sO2 level signature in the blood vessel. In Steps 8–9, we estimated sO2 with a model including SSF and LCA and identified and excluded contaminated B-scans. In Step 10, we adaptively evaluated the effect of SCs on the data by repeating Steps 2–9 with different models of SDBG. The relative contributions of scattering and absorption were accounted for by varying SSF for each of these models. In Steps 11–12, we filtered the

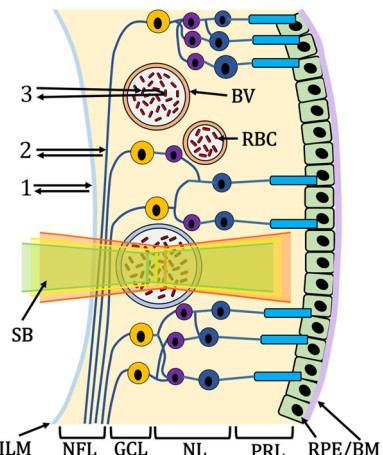

**Fig. 1 Schematic of vis-OCT photon scattering within the retina.** Illustration of the human retina composed of inner-limiting-membrane (ILM), Retinal Nerve Fiber Layer (NFL), blood vessels (BV; red is the artery, blue is the vein), red blood cells (RBC), ganglion cell layer (GCL), nuclear layers (NL) representing the inner nuclear layer to the outer nuclear layer, photoreceptor layers (PRL) containing rods and cones, and the retinal pigment epithelium and Bruch's membrane (RPE/BM). Group 1 highlights the photon path of a specular reflection; 2 highlights the photon path of backscattering without blood attenuation; 3 highlights the photon path of backscattering from red blood cells. A scanning beam (SB) is composed of visible-light wavelengths (green, yellow, and red illustrate different spectral bands of the beam).

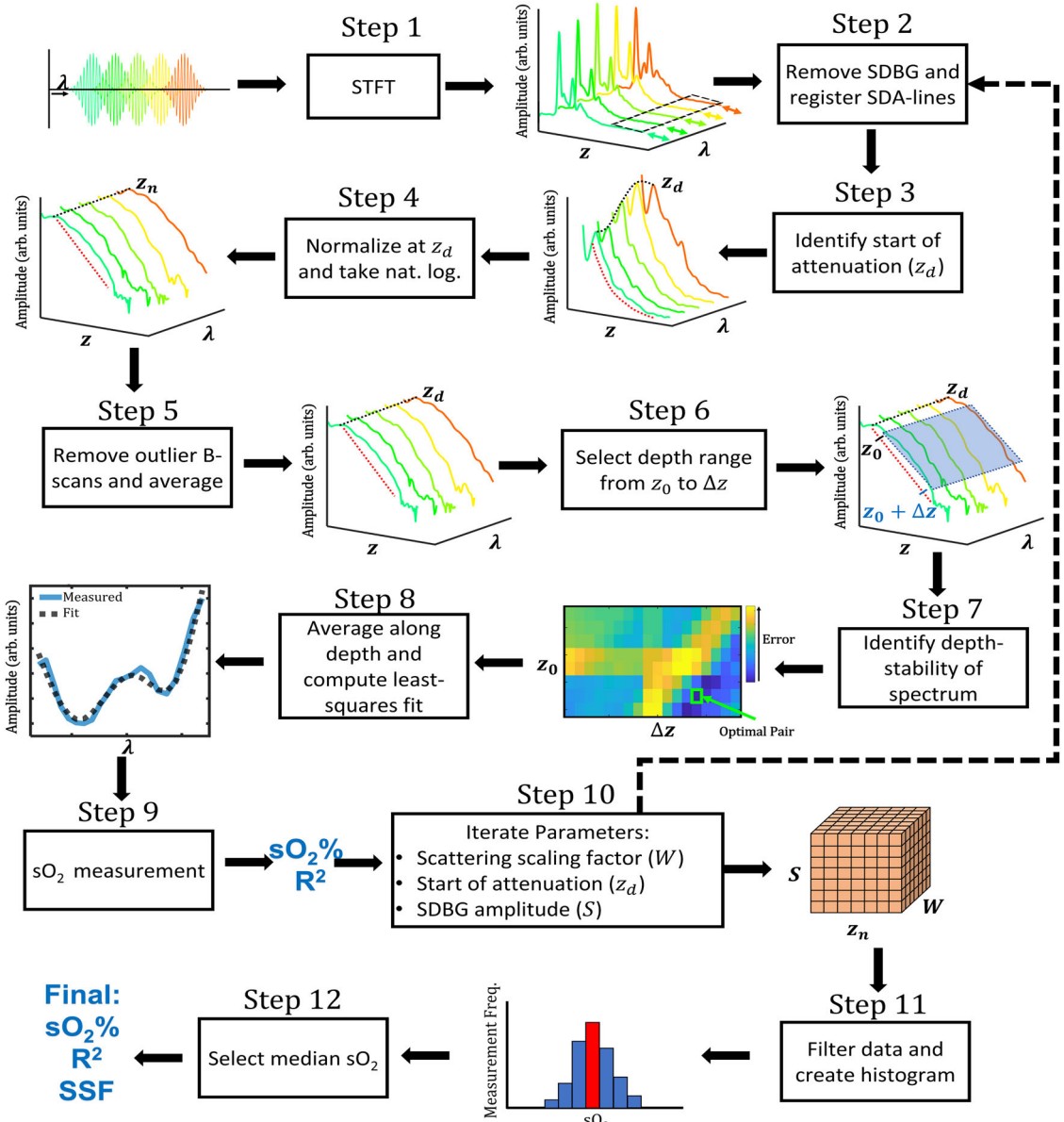

**Fig. 2 Steps and methodology of ADS-vis-OCT.** Flow chart overview of ADS-vis-OCT processing for retinal oximetry. The arrow direction highlights the input and output of each step.

measurements obtained from the different models in Step 10 to obtain an oximetry measurement.

Specifically, in Step 1, we computed the STFT of the full-band interference fringe to obtain 21 spectrally-dependent A-lines (SDA-lines) from 528 nm to 588 nm (see Short-time Fourier Transform in Methods). The outputs were the SDA-lines, which encoded depth ($z$-axis) and wavelength ($\lambda$-axis). For these SDA-lines, the major SCs are LCA, depth-dependent dispersions[29,31] wavelength-dependent shifts along the $z$-axis (colored arrows), and SDBG, which biases the signal along the $z$ and $\lambda$-axes (highlighted by black-dashed box). We processed SDA-lines in vessels for each acquired B-scan.

In Step 2, we axially registered the SDA-lines across all B-scans to account for misalignments caused by motions, dispersion, and LCA. We also removed the SDBG bias in the SDA-lines[32].

In Step 3, we automatically identified the depth ($z_d$) corresponding to the beginning of signal decay in the blood (see Normalization Model for Spectroscopic vis-OCT in Methods). As the Step 3 output, the spectrum at $z_d$ is highlighted by the

black dashed line, and the blood signal decay is highlighted by the red dashed line.

In Step 4, we normalized the SDA-lines by their amplitude at $z_d$ and computed their natural logarithm (see Normalization Model for Spectroscopic vis-OCT in Methods). The Step 4 output is the natural logarithm SDA-lines (NL-SDA-lines). The black dashed line highlights the spectrum at $z_d$, which is constant across $\lambda$ after normalization.

In Step 5, we coarsely measured signal quality in NL-SDA-lines (NL-SDA-lines) in each B-scan (see Fig. S5). Then, after removing outlier B-scans not passing pre-defined quality thresholds, we averaged the remaining B-scans to reduce noise, yielding a single set of NL-SDA-lines.

In Step 6, we selected the depths for spectroscopic measurement. The Step 6 outputs are the starting depth ($z_0$) and depth range ($\Delta z$) for the measurement. The blue box highlights the measurement range for selected depths and spectral sub-bands. We averaged NL-SDA-lines in the blue box along the $z$-axis,

yielding a one-dimensional (1D) STFT spectrum (see Depth Averaging in Methods).

In Step 7, we assessed how the STFT spectrum changed along the depth of the vessel (see Depth Selection in Methods). The Step 7 output is the spectral-stability matrix (SSM), which plots how the spectrum shape changes (error) with depth as a function of $z_0$ (starting depth) and $\Delta z$ (depth range). We selected the optimal pair of $z_0$ and $\Delta z$ with the smallest error (green box), representing the depth region in the vessel where optical attenuation is most constant.

In Step 8, we averaged the NL-SDA-lines within the optimal depth range and fit the STFT spectrum to a linear combination of oxygenated and deoxygenated blood attenuation spectra (see Oximetry Fitting Model in Methods). The Step 8 output is the least-squares regression fit (black dashed line) to the measured STFT spectrum (blue line).

In Step 9, we calculated oxygenation from the regression, outputting sO$_2$ (%) and coefficient of regression ($R^2$).

In Step 10, we repeated steps 2–9 for small variations in three parameters, including scattering scaling factor (SSF), the start of attenuation depth $z_d$, and SDBG amplitude scaling factor $S$ (see Fig. S5). The outputs of Step 10 are 3D matrixes that store the measured sO$_2$ and $R^2$, respectively, where each dimension corresponds to each of the three parameters.

In Step 11, we organized and filtered the data from each iteration (see Supplementary Methods 1).

In Step 12, we selected the median sO$_2$ value from the distribution (red bin in the histogram) as the final output and its corresponding $R^2$ and SSF values.

**Short-time Fourier transform**. We multiplied 21 Gaussian windows with the spectral interferogram. Windows were of equal wavenumber ($k$) full-width at half maximum (FWHM) and spaced equidistantly in $k$ space from 528 nm to 588 nm. Window FWHM was 11 nm at 558 nm, reducing the average axial resolution to 9 μm in the retina (assuming a refractive index of 1.35).

**Normalization model for spectroscopic vis-OCT**. To accurately quantify sO$_2$, the oxygen-dependent attenuation spectrum of blood must be isolated from sample-dependent and system-dependent SCs. Equation (1) describes the generalized SDA-line for a homogenous medium

$$I(\lambda, z) = F(\lambda, z)2\sqrt{I_{samp}(\lambda)I_{ref}(\lambda)}\sqrt{A(\lambda, z)T(\lambda, z_s)\mu_b(\lambda)}e^{-\mu_t(\lambda)(z-z_s)} + B(\lambda, z),$$ (1)

where $\lambda$ is the wavelength; z is the depth coordinate; $z_s$ is the surface of the medium with respect to the zero-delay depth z = 0; $I_{samp}(\lambda)$ and $I_{ref}(\lambda)$ are the power spectra of the light collected from the sample and reference arms, respectively. The SDR is $F(\lambda, z)$[25], the LCA transfer function is $A(\lambda, z)$ (see Fig. S4), and the SDBG is $B(\lambda, z)$[26], $\mu_b(\lambda)$ is backscattering coefficient of the medium, $\mu_t(\lambda)$ is the attenuation coefficient of the medium, and $T(\lambda, z_s)$ is the double-pass transmission coefficient across the top interface of the medium

The retina-specific model for the SDA-line must consider multi-layered media with different optical properties. After normalizing by the source power spectrum (DC component) and subtracting the SDBG, we write the amplitude of the SDA-line where blood signal decay begins, $z_d$ (Step 3 in Fig. 2), as

$$I(\lambda, z_d) = F(\lambda, z_d)2\sqrt{A(\lambda, z_d)\mu_{b_{blood}}(\lambda)}$$
$$\prod_{i=1}^{n-1}\left[\sqrt{T(\lambda, z_i)}e^{-\mu_{t_i}(\lambda)(z_{i+1}-z_i)}\right] for\ z = z_d,$$ (2)

where $i$ is the tissue layer and blood is the $n^{th}$ tissue layer. We write the residual SDA-line below $z_d$ as

$$I(\lambda, z) = I(\lambda, z_d)\frac{F(\lambda, z)\sqrt{A(\lambda, z)}}{F(\lambda, z_d)\sqrt{A(\lambda, z_d)}}e^{-\mu_{t_{blood}}(\lambda)(z-z_d)} for\ z > z_d,$$ (3)

where $\frac{F(\lambda, z)\sqrt{A(\lambda, z)}}{F(\lambda, z_d)\sqrt{A(\lambda, z_d)}}$ represents the residual LCA and SDR beyond the depth $z_d$. We divided $I(\lambda, z)$ by $I(\lambda, z_d)$ to yield

$$I(\lambda, z) = \frac{F(\lambda, z)\sqrt{A(\lambda, z)}}{F(\lambda, z_d)\sqrt{A(\lambda, z_d)}}e^{-\mu_{t_{blood}}(\lambda)(z-z_d)} for\ z > z_d.$$ (4)

We rejected all vessels from depths greater than 700 μm from the zero-delay position. We calculated $\frac{F(\lambda, z)}{F(\lambda, z_d)}$ from the roll-offs of our spectrometer and found the SDR had negligible spectral influence after normalization by $I(\lambda, z_d)$. Therefore, the ratio $\frac{F(\lambda, z)}{F(\lambda, z_d)}$ is set to 1, yielding

$$I(\lambda, z) = \frac{\sqrt{A(\lambda, z)}}{\sqrt{A(\lambda, z_d)}}e^{-\mu_{t_{blood}}(\lambda)(z-z_d)} for\ z > z_d.$$ (5)

We estimated $\frac{\sqrt{A(\lambda, z)}}{\sqrt{A(\lambda, z_d)}}$ for the same depths (see Fig. S4) and concluded that the residual LCA could have a small but non-negligible influence on sO$_2$ even after normalization. Therefore, we included $\frac{\sqrt{A(\lambda, z)}}{\sqrt{A(\lambda, z_d)}}$ in our model. Finally, taking the natural logarithm of $I(\lambda, z)$, we have a function that is linearly proportional to $\mu_{t_{blood}}(\lambda)$ (Fig. 2, Step 3) as

$$\ln(I(\lambda, z)) = \frac{1}{2}(\ln(A(\lambda, z)) - \ln(A(\lambda, z_d))) - \mu_{t_{blood}}(\lambda)(z - z_d) for\ z > z_d.$$ (6)

**Depth averaging**. To remove noise variations in the speckle and the background, we averaged Eq. (6) across $z$, which can be written as

$$S(\lambda, z_o, \Delta z) = <\ln(I(\lambda, z))>_z = \frac{1}{\Delta z}\int_{z_0}^{z_0+\Delta z}\ln(I(\lambda, z))dz$$

$$= \frac{1}{\Delta z}\int_{z_0}^{z_0+\Delta z}\frac{1}{2}(\ln(A(\lambda, z) - A(\lambda, z_d)))$$
$$- \mu_{t_{blood}}(\lambda)(z - z_d)dz\ for\ z > z_d = LCA_{avg}(\lambda, z)$$
$$- \mu_{t_{blood}}(\lambda)\left[(z_0 - z_d) + \frac{\Delta z}{2}\right] for\ z > z_d,$$ (7)

where $LCA_{avg}$ is the depth-averaged residual LCA (Fig. S4); $z_0$ is the starting depth for averaging; and $\Delta z$ is the depth range for averaging. We show the statistical advantages of depth averaging in Table S3.

**Depth selection**. The term $-\mu_{t_{blood}}(\lambda)\left[(z_0 - z_d) + \frac{\Delta z}{2}\right]$ is a constant and represents uniform, oxygen-dependent attenuation along the vessel depth. Blood cell packing, orientation, flow, and specifically laminar flow in venules, and oxygen diffusion may add variability to this assumption. To reduce variability and outliers in sO$_2$ measurement, we minimized depth-dependent changes to the attenuation spectrum.

First, we measured the STFT spectrum in a blood vessel according to Eq. (7). We iterated $z_0$ from 0 to 12 $\mu m$ and $\Delta z$ from 17 to 40 $\mu m$, both in 1.15 $\mu m$ (depth pixel size) increments. We normalized each spectrum to a minimum of 0 and a maximum of 1. Then, we generated a 3D matrix that indexed the spectra according to their respective depth windows. Such a matrix can be written as

$$S(\lambda_k)_{m,n} = \begin{pmatrix} S(\lambda_k)_{1,1} & \cdots & S(\lambda_k)_{1,20} \\ \vdots & \ddots & \vdots \\ S(\lambda_k)_{10,1} & \cdots & S(\lambda_k)_{1,20} \end{pmatrix}, \quad (8)$$

where $S(\lambda_k)_{m,n}$ is the normalized (between 0 and 1) spectrum for each window iteration; $m$ is the iteration index of $z_0$; $n$ is the iteration index of $\Delta z$; and $k$ is the STFT sub-band index. To measure the response of $S(\lambda_k)_{m,n}$ to a shift in depth, we computed the mean-squared-error (MSE) between spectra from 9 adjacent windows in $S(\lambda_k)_{m,n}$ to generate the SSM as

$$SSM_{i,j} = \sum_{k=1}^{21} \sum_{x=-1}^{1} \sum_{y=-1}^{1} \sum_{m=2}^{9} \sum_{n=2}^{19} \left( MSE\left[ S(\lambda_k)_{m,n}, S(\lambda_k)_{m+x,n+y} \right] \right), \quad (9)$$

where $MSE\left[ S(\lambda_k)_{m,n}, S(\lambda_k)_{m+x,n+y} \right]$ is the MSE between spectra $S(\lambda_k)_{m,n}$ and $S(\lambda_k)_{m+x,n+y}$; and $x$ and $y$ are the indexes of the compared spectra. We show an example $SSM_{m,n}$ from two selected vessels in Fig. S6. We identified the indexes $m$ and $n$ for minimized $SSM_{m,n}$ and used this depth window for $sO_2$ measurement.

**Oximetry fitting model**. We fit the spectrum determined by Eqs. (7–9) to the following model using a non-negative linear least-squares regression[33]

$$\begin{aligned} S(\lambda, C_{HbO_2}, C_{Hb}, SSF) =& LCA_{avg}(\lambda, z) - (C_{HbO_2}(\mu_{a_{HbO_2}}(\lambda) \\ &+ SSF\mu_{s_{HbO_2}}(\lambda)) + (C_{Hb}(\mu_{a_{Hb}}(\lambda) \\ &+ SSF\mu_{s_{Hb}}(\lambda))) \left[ (z_0 - z_d) + \frac{\Delta z}{2} \right] for\ z > z_d, \end{aligned} \quad (10)$$

where $\mu_{a_{HbO_2}}(\lambda)$, $\mu_{s_{HbO_2}}(\lambda)$, $\mu_{a_{Hb}}(\lambda)$, and $\mu_{s_{Hb}}(\lambda)$ are the reported absorption and scattering coefficients of oxygenated and deoxygenated blood, respectively, and SSF is a scattering scaling factor. After fitting the spectrum, we measure $sO_2 = \frac{C_{HbO_2}}{C_{HbO_2}+C_{Hb}}$, which is constrained between 0 and 1 due to the nature of the non-negative linear regression. We constrained the fitting for $SSF = 0.02 - 0.10$, consistent with our previous findings[34] and $\pm 100\ \mu m$ focal shifts in $LCA_{avg}(\lambda, z)$ (see Fig. S4).

We used the reported absorption and Mie-theory-predicted scattering coefficients of blood[13,22]. We modified the reported spectra to match the post-processing of the vis-OCT signal. First, we cropped wavelengths from the reported spectra within our spectrometer's wavelength range (508–614 nm). Next, we upsampled the reported spectra by 6-fold to a 12288-element array, the same size as our interference fringe after 6-fold zero-padding. Then, we performed interpolation of the reported spectra to be linear in $k$-space. Finally, we filtered and digitized the reported blood spectra with the same 21 STFT Gaussian windows.

**Vis-OCT systems**. We used vis-OCT systems at NYU Langone Health Center (Aurora X1, Opticent Health, Evanston, IL) and Northwestern Medical Hospital (Laboratory Prototype), which

were reported, respectively, in our previous work[30]. We previously characterized spectrometer roll-offs[25].

**Imaging protocols**. Imaging was performed at NYU Langone Health Center and Northwestern Memorial Hospital. We limited light exposure on the cornea to <250 $\mu W$, which is considered eye-safe[35]. The camera line period was set to 40 $\mu s$ (39 $\mu s$ exposure + 1 $\mu s$ data transfer) for a 25 kHz A-line rate. All imaging was approved by respective NYU (study number i16-01302_CR6) and Northwestern University (study number STU00215100) Institutional Review Boards and adhered to the Tenants of Helsinki. We advertised the study among researchers at NYU Ophthalmology and Northwestern University Biomedical Engineering departments. All participants provided either written or oral consent after being informed of the goals of the intended study, maximum image acquisition time, laser safety, and protection of personal identity.

We performed three scan types for oximetry measurement: arc scan, small FOV raster scan, and large FOV raster scan. An arc scan is a 120-degree segment of a circular scan with a radius of 1.7 mm acquired with 16 B-scans at 8192 A-lines per B-scan. A small-FOV scan is a 1 mm × 1 mm raster scan acquired containing 16 B-scans with 8192 A-lines per B-scan. A large-FOV scan is a 4.8 mm × 4.8 mm raster scan containing 64 B-scans with 4096 A-lines per B-scan. We found no significant differences in $sO_2$ values for different scanning modes.

**Research participants demographics**. The 18 research participants were between the ages of 21 and 62 with an average age of 38.1 and median age of 33.1. Of the 18 research participants, 9 were female, 9 were male, 16 were white, and two were Asian. All research participants were recruited from NYU and Northwestern University research labs and had no known ocular or systematic diseases. For the 16 white research participants, eight were female, eight were male, and the average age was 39.2. For the two Asian research participants, one was female, one was male, and the average age was 29. Of all research participants, six were between the ages of 20–29, five were between the ages of 30–39, three were between the ages of 40–49, one was between the ages of 50–59, and three were between the ages of 60–62.

**Vessel selection**. We measured $sO_2$ in 175 total retinal vessels (100 arteries and 75 veins) across 18 healthy research participants. We analyzed 125 unique retinal vessels (72 arteries and 53 veins) (see Methods—Retinal oximetry in a healthy cohort). For vessels with more than one measurement, we selected unique vessels by selecting $sO_2$ measurement with the highest $R^2$. For repeatability analysis, we selected vessels with at least two repetitions.

**Vessel segmentation**. We selected the left and right borders of a vessel, guided by its attenuation shadow. To account for different vessel geometries, we automatically segmented the central 36%, 40%, and 42% of A-lines of the vessel. We repeated these three segmentations for a 4% shift left and right of the detected vessel center, totaling 9 segmentations of the same vessels. We treated each of the 9 segmentations as separate B-scans in the analysis.

**Statistics and reproducibility**. To assess whether the lower $sO_2$ values were influenced by vessel diameter or were an artifact of poor fitting (lower $R^2$), we included both parameters in a linear model

$$sO_2 = a_1 D + a_2 R^2, \quad (11)$$

where $D$ is vessel diameter; $R^2$ is the spectral fit $R^2$ value; and $a_1$ and $a_2$ are arbitrary constants. A two-way ANOVA was

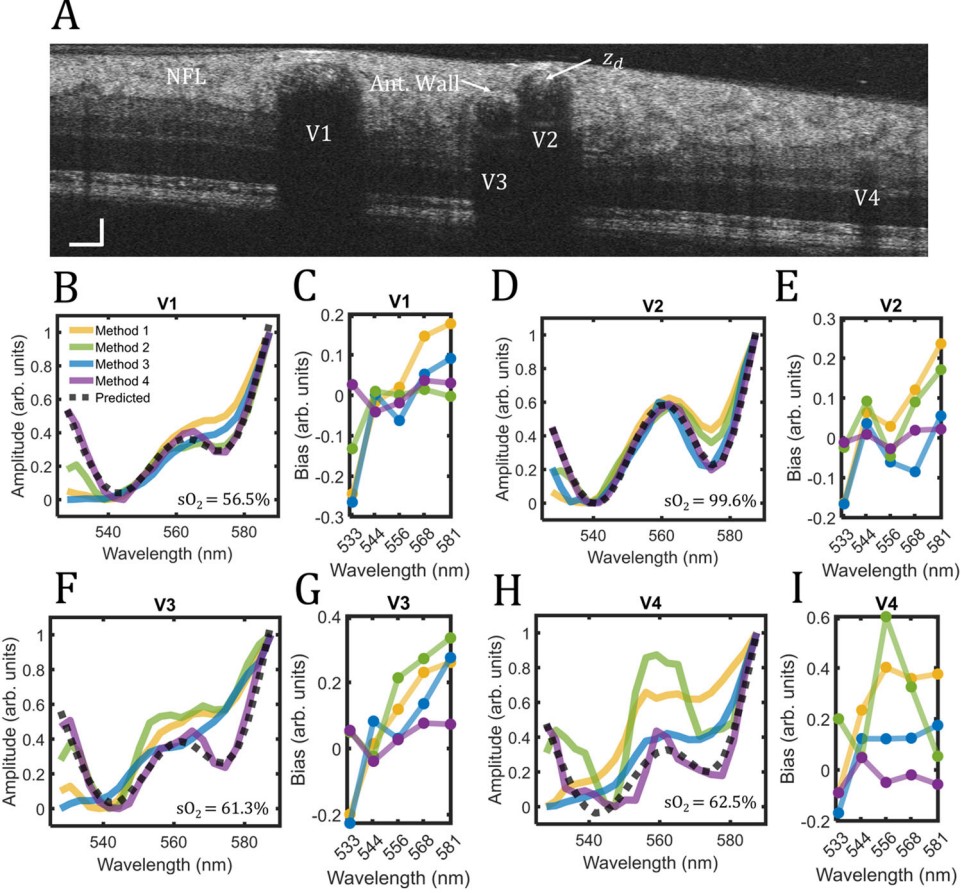

**Fig. 3 Comparison of spectroscopic normalization methods.** Spectroscopic normalizations in the human retina. **A** vis-OCT B-scan of a healthy human retina; vessels labeled V1-V4 were identified as vein, artery, vein, and vein, respectively; **B** Measured spectra in V1 with no normalization (yellow), normalization by NFL (green), normalization by the anterior vessel wall (blue), and normalization by the start of signal decay in the blood ($z_d$) (purple), respectively; predicted spectrum for $sO_2 = 56.5\%$ (gray dashed line); **C** Bias for each normalization with respect to predicted spectrum at five wavelengths; **D**, **E** Same analysis for V2 and $sO_2 = 99.6\%$; **F**, **G** Same analysis for V3 and $sO_2 = 61.3\%$; **H**, **I** Same analysis for V4 and $sO_2 = 62.5\%$.

performed in MATLAB 2018a. Significance was considered as $p<0.05$. We compared $sO_2$ measurement populations in "Retinal oximetry in a healthy volunteer group". We used the two-sample t-test to determine differences in the mean. Significance was considered as $p<0.05$. t-tests were performed in MATLAB 2018a. Our dataset includes 125 unique vessels from 18 volunteers. In total, measurements were taken from 72 arteries (36 larger arteries and 36 smaller arteries) and 53 veins.

**Reporting summary**. Further information on research design is available in the Nature Portfolio Reporting Summary linked to this article.

## Results

**Optimal tissue normalization**. In ADS-vis-OCT Step 4, we normalize SDA-lines using a reference signal in the tissue to remove SCs. However, there is no consensus in the literature on the optimal location for normalization or even if it is necessary. Here, we highlight four potential tissue normalizations for vis-OCT oximetry: Method 1: no normalization, as reported by refs. [16,20] Method 2: normalization by the nerve fiber layer (NFL), which is typically anterior to the retinal vessels, as reported by ref. [24]. and suggested by refs. [18,19]. Method 3: normalization by the anterior vessel wall, which can be highly reflective and is immediately above the blood signal; and Method 4: normalization at the start of signal decay in the blood ($z_d$), which is used in ADS-vis-OCT. We compare the influence of

Methods 1–3 with Method 4 on the STFT spectra in human retinal vessels. For a fair comparison, we use the same SDA-lines and vessel measurement depths for all methods.

Figure 3A is a vis-OCT B-scan image acquired 1.7-mm superonasal to the optic disc from a 37-year-old volunteer. We measure STFT spectra in vessels V1-V4 with diameters of 168 μm, 120 μm, 79 μm, and 43 μm, respectively. Figure 3B plots the measured STFT spectra in V1 for all four normalization methods (yellow, green, blue, and purple, respectively). The gray dashed line plots the literature-derived spectrum (see Methods—Oximetry fitting model) predicted using least-squares fit in ADS-vis-OCT ($sO_2 = 56.5\%$). Since we could not guarantee that Methods 1–3 completely removed SCs, it is mathematically inappropriate to fit their STFT spectra with the ADS-vis-OCT fitting model. Instead, we use the ADS-vis-OCT predicted spectrum (gray dashed line) as a reference with which to identify SC-derived errors. We scale all STFT spectra between 0 and 1 in the plots.

Comparing the normalizations, Methods 1–3 measure STFT spectra that deviate significantly from the predicted spectrum, while Method 4 agrees with the predicted spectrum. We quantify the general trends of the spectral biases from Fig. 3B to Fig. 3C for five STFT bands with their respective central wavelengths at 533 nm, 544 nm, 556 nm, 568 nm, and 581 nm. The bias is defined as the difference between each measured spectrum in Fig. 3B and the predicted spectrum for the specified central wavelengths of the STFT bands. The bias trends for Methods 1–4 highlight different error-inducing SCs, confirming the observation of different spectral

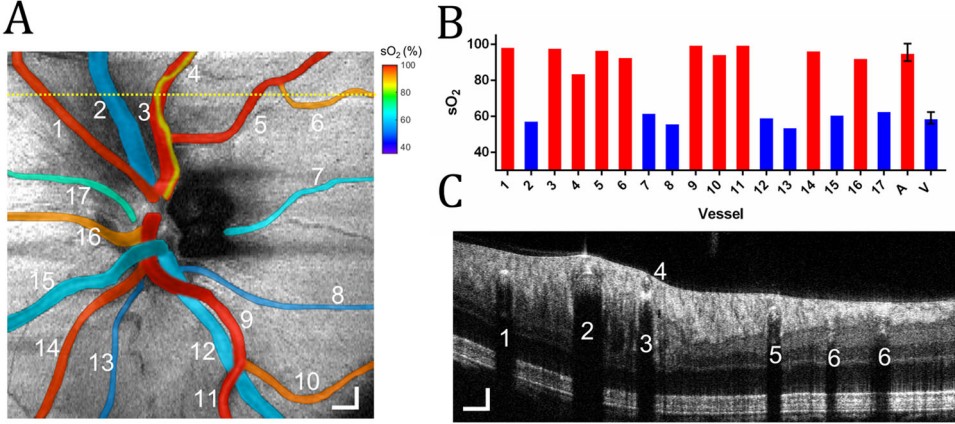

**Fig. 4 Oximetry map of the optic disk. A** sO$_2$ measurements in 17 vessels in the optic disk from a healthy 23-year-old. The sO$_2$ values pseudo-colored and overlaid onto the fundus image. Scale bar: 300 µm; **B** Bar chart plots sO$_2$ measurements from panel A in individual arteries (red bar) and veins (blue) numbered from 1 to 17, as well as average sO$_2$ in all arteries and all veins; **C** B-scan from the position highlighted by the yellow dashed line in (**A**.) Scale bars: 150 µm.

features in Fig. 3B. Although Methods 1-3 induce larger bias values for the longest wavelengths (581 nm) versus the shortest wavelengths (533 nm), their trends are non-trivial (e.g., non-linear or non-monotonic) with different amplitudes and signs. Such biases may represent either incomplete removal of SCs or the introduction of new SCs after tissue normalization.

Figure 3D, E, F, G, and H, I shows the same analysis for vessels V2, V3, and V4, respectively. Similar to vessel V1 (Fig. 3B, C), biases in vessel V2 (Fig. 3D, E) for Methods 1–3 are generally higher at longer wavelengths (581 nm) than for shorter wavelengths (533 nm) but differ at middle wavelengths (544–568 nm). Also, compared with V1, V2 bias magnitudes are generally increased for Methods 1 and 2 but reduced for Method 3. The unpredictability of SCs became particularly noticeable in vessels V3 & V4, which are smaller than V1 and V2 and buried deeper below the NFL. Vessel V3 (Fig. 3F, G) has nearly double the bias amplitudes compared with V1 and V2. Also, Method 2 bias is highest in the middle through longer wavelengths (556–581 nm), a trend not seen in V1 and V2. Finally, vessel V4 demonstrates the most significant biases among all the vessels, with a particularly large bias at the middle wavelength (556 mm).

Figure 3 indicates that normalization Methods 1–3 induce wavelength-dependent biases that vary among vessels and are unpredictable. Method 4 used in ADS-vis-OCT consistently yields the least bias across the entire spectral range and agrees well with the predicted model. Observing that SCs vary with retinal locations, eye anatomies, and imaging sessions makes proper normalization essential for accurate and repeatable spectroscopic analysis.

**Retinal oximetry around the optic disk**. An oximetry map of the optic disk can help investigate oxygen delivery or extraction in the entire retina. Figure 4A shows a representative vis-OCT oximetry map of the optic disk with a 4.8 × 4.8 mm$^2$ FOV in a 23-year-old volunteer. We measure sO$_2$ in 17 vessels (10 arteries and seven veins) ranging from 37 µm to 168 µm in diameter.

We pseudo-color the 17 vessels according to their measured sO$_2$ values onto a vis-OCT fundus image (Fig. 4A) and plot the sO$_2$ values in the bar chart (Fig. 4B). The measured sO$_2$ across all arteries is 95.8 ± 4.4% ($n = 10$) and the measured sO$_2$ across major arteries (diameter $\geq$ 100 µm) is 97.3 ± 2.8% ($n = 6$). The average pulse oximeter measurement from the index finger is 98%, agreeing well with the vis-OCT measured sO$_2$ from the major arteries. The measured sO$_2$ across all veins is 59.0 ± 3.2%.

Figure 4C shows a B-scan from the location highlighted by the dashed yellow line in Fig. 4A. We can observe a small artery (vessel 4, diameter = 37 µm) directly above a major artery (vessel 3, diameter = 122 µm). The measured sO$_2$ value in vessel 3 is 98.3%, consistent with the pulse oximeter reading (98%); the measured sO$_2$ value in vessel 4 is 85.8%. We measured sO$_2$ values from both vessels, demonstrating the unique depth-resolved sO$_2$ imaging capability permitted by vis-OCT. Axially overlapping vessels in the retina require SC removal since, for example, attenuation from vessel 4 will contaminate that in vessel 3. Vessel 3's agreement with the pulse oximeter, despite sitting below a vessel with a significantly lower sO$_2$, confirms the removal of SCs from posterior tissues in vivo. The posterior wall of vessel 4 appears to be in direct contact with the anterior wall of vessel 3. The exact reason for the lower sO$_2$ in vessel 4 and whether this is a common phenomenon for a smaller blood vessel connecting to a larger artery is unknown and requires further investigation.

**Retinal oximetry in a healthy volunteer group**. We perform ADS-vis-OCT retinal oximetry in 18 volunteers without known ocular diseases. We measure sO$_2$ in 125 unique vessels (72 arteries and 53 veins) within a 3.4 mm radius of the optic nerve head.

Figure 5A shows sO$_2$ from unique arteries (red) and veins (blue) plotted as a function of vessel diameter. Arterial sO$_2$ values show a decreasing trend with decreasing vessel diameter. We determine that vessel diameter is a statistically significant factor (see Methods—Statistical Analysis) in this trend ($p = 4.35 \times 10^{-6}$). The diameter-dependent trend is consistent with oxygen gradients observed in other precapillary arteries[36–42]. Since smaller vessels generally offers less attenuation contrast and fewer-averaged pixels than larger vessels, they are potentially more sensitive to noise. We investigate whether the sO$_2$ decrease is an artifact of lower spectral fit $R^2$. We determine that $R^2$ is not a statistically significant factor (see Methods—Statistical Analysis) in this trend ($p = 0.701$). Finally, venous sO$_2$ slightly increases with decreasing vessel diameter, but the trend is not statistically significant ($p = 0.232$). Spectral fit $R^2$ is also not significant ($p = 0.070$) in determining sO$_2$ in veins.

To account for the observed sO$_2$ gradient with diameter, we compute average sO$_2$ in arteries across two diameter groups (diameter $\geq$ 100 µm and diameter <100 µm). Figure 5B shows sO$_2$ measurements for major arteries (diameter $\geq$ 100 µm), small arteries (diameter < 100 µm), and veins with all diameters. Major arteries ($n = 36$) has sO$_2$ = 97.9 ± 2.9%. Small arteries ($n = 36$)

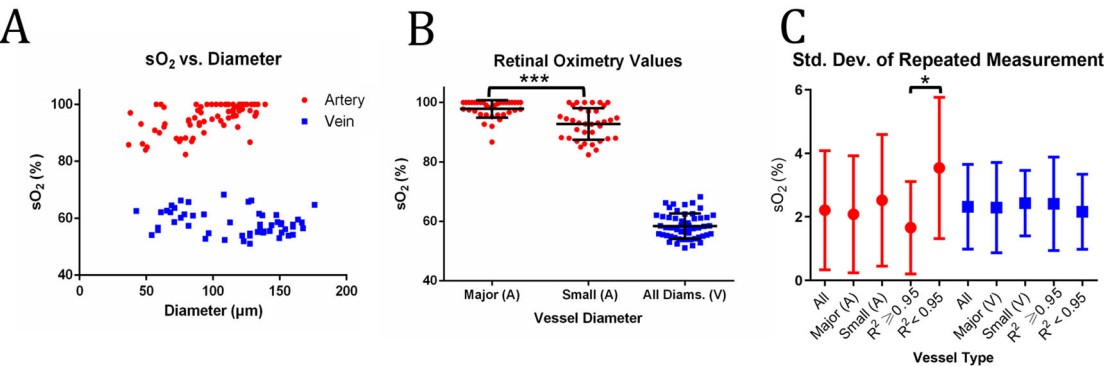

**Fig. 5 Retinal sO₂ in a healthy cohort. A** sO₂ measurements in 72 unique arteries (red) and 53 unique veins (blue) plotted against vessel diameter from 18 healthy volunteers; **B** Distribution of sO₂ measurements for major (diameter ≥ 100 μm, n = 36) and small (diameter < 100 μm, n = 36) artery calibers and all veins; **C** Repeatability of arteries and veins (n = 24 for all artery, n = 17 for major artery, n = 7 for small artery, n = 17 for artery with $R^2$ ≥ 0.95, n = 7 for artery with $R^2$ < 0.95, n = 18 for all vein, n = 15 for major vein, n = 3 for small vein, n = 11 for vein with $R^2$ ≥ 0.95, n = 7 for vein with $R^2$ < 0.95). * indicates p <0.05 and *** indicates p <0.001 from two-sample t-test. All error bars are standard deviations.

has sO₂ = 93.2 ± 5.0%. The difference in sO₂ between the two groups is statistically significant ($p = 4.01 \times 10^{-6}$). Average spectral fits are $R^2 = 0.96$, 0.93, and 0.95 for major arteries, small arteries, and all veins, respectively.

We acquire repeated scans of 42 unique vessels and calculate their average standard deviations (SDs) (24 unique arteries and 18 unique veins across 12 volunteers, see Table S1). Figure 5C shows average SDs for unique arteries (red) and veins (blue). All arteries and veins have average SDs of 2.2% and 2.3%, respectively. First, we investigate repeatability for arteries and veins of diameters larger and smaller than 100 μm. Larger arteries (n = 17) and smaller arteries (n = 7) have average SDs of 2.1% and 2.5%, respectively. Larger veins (n = 15) and smaller veins (n = 3) have average SDs of 2.3% and 2.4%, respectively. There is no statistically significant difference between the average repeatability values for any of the groups (two-sample T-test). Finally, we investigate repeatability for best spectral fits ($R^2 \geq 0.95$) and relatively lower spectral fits ($R^2 < 0.95$). Best fit arteries (n = 17) and lower-fit arteries (n = 7) have average SDs of 1.7% and 3.5%, respectively. Best-fit veins (n = 11) and lower-fit veins (n = 7) have average standard deviations of 2.4% and 2.2%, respectively. The difference between the average SDs for the artery groups is statistically significant ($p = 0.02$), although one analyzed artery with $R^2 < 0.95$ is an outlier (SD = 6.83%). Without the outlier, the average SD $R^2 < 0.95$ arteries is 3.0%, not a statistically significant difference ($p = 0.09$). Statistics for each unique vessel in repeatability analysis are shown in Fig. S2.

We estimate the ground-truth arterial sO₂ using a pulse oximeter fastened to the finger during imaging and compare it with ADS-vis-OCT measured sO₂ in major arteries (diameter ≥ 100 um) from 12 research participants. The average ADS-vis-OCT sO₂ in research participants is 98.3%, in very close agreement with that from the pulse oximeter (sO₂ = 98.6%). We measure the root-mean-squared-error (RMSE) between pulse oximeter sO₂ and ADS-vis-OCT sO₂ for each unique artery in each respective research participant (see Table S2). The average RMSE between vis-OCT and the pulse oximeter is 2.1%, the same value as the SD of repeated measurements in major arteries. This suggests that ADS-vis-OCT measured sO₂ may be within noise-limited agreement with the pulse oximeter.

**Comparison with non-adaptive retinal sO₂ measurements.** To quantify the influence of adaptive processing (Fig. 2) on sO₂, we compare measurements using non-adaptive vis-OCT methods. Briefly, we define non-adaptive processing as using fixed algorithms or parameters. To achieve a non-adaptive version of ADS-vis-OCT, we omit Steps 3, 5, 7, 10, 11, and 12 from the flow chart in Fig. 2,

which eliminates the adaptive signal normalization, adaptive quality control, adaptive depth selection, and parameter iterations (see Supplementary Methods 1 for more detail). The above algorithm is a singular measurement of blood's attenuation coefficient along a pre-defined depth range, which is the basis of all retinal oximetry measurements. Hereon, we refer to this as the "fixed attenuation method" (FA method). Next, we compare ADS-vis-OCT with the "posterior wall" (PW method) reported previously by ref. [16].

Table 1 compares sO₂ measurements in the same 125 unique vessels (Fig. 5) using the FA method, PW method, and ADS-vis-OCT. Unlike ADS-vis-OCT, the FA and PW methods yield a handful of extreme outliers (>7 SDs from the ADS-vis-OCT mean). To highlight the failed measurements while reducing statistical bias, we report the success rate, defined as the percent of physiologically viable measurements from each group of vessels. Additionally, the PW method also fails when there is insufficient visibility of the posterior wall. Empirically, we determine the posterior wall is visible when its average STFT intensity is at least 1 dB greater than the noise floor; otherwise, the measurement is considered unviable.

Using the FA method, we successfully measure most vessels (≥ 85% success). This yields mean sO₂ values of 97.5 ± 5.2% for major arteries, 86.5 ± 10.9% for small arteries, and 52.6 ± 12.5% for all veins. Compared with ADS-vis-OCT, the FA method measures similar sO₂ for larger arteries, (97.5% vs. 97.9%), but underestimates by 7% for small arteries (86.5% vs. 93.2%) and by 6% for all veins (52.6% vs 58.4%). SDs of each vessel group increase 2–3-fold using the FA method, as compared with ADS-vis-OCT. Individual vessel repeatability is similar for major arteries between the two methods (2.3% and 2.1%) but is nearly 4-fold higher for small arteries (9.1% and 2.5%) and veins (8.9% and 2.3%). The less stable measurements are consistent with their worse spectral fits ($R^2$). In particular, small arteries and veins have $R^2$ of 0.86 and 0.87 using FA method, lower than 0.93 and 0.95 for the same ADS-vis-OCT groups.

The PW method fails to perform viable sO₂ measurements in half of the vessels. Specifically, most major arteries and veins fail due to their larger vessel diameter and insufficient visualization of the posterior wall. PW-measured major arteries yielded an sO₂ of 90.7 ± 7.5%, with 6.0% repeatability. These sO₂ measurements are lower and less stable than in ADS-vis-OCT (97.9 ± 2.9%, with 2.1% repeatability). The mean sO₂ of 90.7% is inconsistent with pulse oximeter readings (e.g., sO₂ ≥ 95%). Smaller arteries yielded 92% successful measurement but have 4% sO₂ lower than in ADS-vis-OCT (89.0% and 93.2%), have a broader distribution (SD = 12.8% vs. 5.0%), and are less repeatable (7.4% and 2.5%). Additionally, the mean sO₂ is almost the same between PW-

**Table 1 Comparison of different vis-OCT oximetry methods.**

| Vessel Type | Method | Adaptive | Success % | Mean sO$_2$ (%) | SD sO$_2$ (%) | Ves. Rep. (SD %) | Fit $R^2$ |
|---|---|---|---|---|---|---|---|
| Major Artery ($n = 32$) | FA | No | 89 | 97.5 | 5.2 | 2.3 ($n = 15$) | 0.92 |
| Small Artery ($n = 34$) | FA | No | 94 | 86.5 | 10.9 | 9.1 ($n = 7$) | 0.86 |
| Vein ($n = 45$) | FA | No | 85 | 52.6 | 12.5 | 8.9 ($n = 15$) | 0.87 |
| Major Artery ($n = 13$) | PW | No | 36 | 90.7 | 7.5 | 6.0 ($n = 3$) | 0.94 |
| Small Artery ($n = 33$) | PW | No | 92 | 89.0 | 12.8 | 7.4 ($n = 6$) | 0.95 |
| Vein ($n = 17$) | PW | No | 32 | 62.4 | 13.3 | Not Enough Data | 0.94 |
| Major Artery ($n = 36$) | ADS | Yes | 100 | 97.9 | 2.9 | 2.1 ($n = 17$) | 0.96 |
| Small Artery ($n = 36$) | ADS | Yes | 100 | 93.2 | 5.0 | 2.5 ($n = 7$) | 0.93 |
| Vein ($n = 53$) | ADS | Yes | 100 | 58.4 | 4.3 | 2.3 ($n = 18$) | 0.95 |

Comparison of adaptive and non-adaptive vis-OCT oximetry; n is the number of vessels analyzed for each group; FA is the fixed attenuation method; PW is the posterior wall method; Ves. Rep. is the repeatability of unique vessels as defined in Results – Retinal Oximetry in a Healthy Population.

measured major and small arteries, meaning it fails to detect the statistically significant difference measured by ADS-vis-OCT. Finally, all veins measured with the PW method have 4% higher sO$_2$ than ADS-vis-OCT with a broader distribution (SD = 13.3% vs. 4.3%) (insufficient data to measure repeatability). Interestingly, the PW method has higher spectral fits ($R^2$) than the FA method, comparable to those in ADS-vis-OCT. However, PW measurements also have significantly increased uncertainties, possibly caused by the sO$_2$ fitting model[16]. Since the PW method does not correctly remove SCs, it can misinterpret SCs as true sO$_2$-carrying signals. In this case, a high $R^2$ with high measurement uncertainty indicates the model is overfitting the data. Such overfitting can lead to systemic biases, and uncertainties intrinsic to the least-squares fit, as evident in Table 1. On the contrary, the FA method and ADS-vis-OCT directly correlate increased uncertainty with lower $R^2$ fit using the same regression model developed in this work (see Oximetry fitting model in Methods).

## Discussion

Existing vis-OCT oximetry shows limited repeatability caused by SCs and uniformly applied algorithms to all vessels. To overcome this challenge, we developed a model for vis-OCT oximetry and an adaptive processing method that optimizes SC removal and sO$_2$ measurement. In the human retina, sO$_2$ measurements across all unique arteries and veins were highly repeatable (SD = 2.2% and 2.3%, respectively). Furthermore, the RSME of 2.1% between major artery sO$_2$ and a pulse oximeter was the same as the average SD of major artery sO$_2$ after repeated measurements (SD = 2.1%), suggesting that uncertainty was limited by random noise, not systemic bias. Compared with pulse oximetry, existing studies using fundus photography-based retinal oximetry found the sO$_2$ values in major retinal arteries to be 2–5% lower without statistical significance, where the authors attributed such discrepancies to calibration errors or diffusion[43–45]. Using ADS-Vis-OCT, we measured an average sO$_2$ of 98.3% in major arteries and 93.2% in smaller arteries, which are slightly lower than the mean pulse oximeter value of 98.6%.

Across 72 unique arteries from 18 research participants, we found a statistically significant trend between decreased diameter and decreased sO$_2$. This trend is consistent with previously observed precapillary oxygen gradients[36–42]. Across measured arteries and veins, vessel diameter ranged from 37 μm to 176 μm. We excluded vessels below a diameter of 35 μm as backscattering becomes heavily orientation-dependent along the direction of the laminar RBC flow. This is amplified in capillaries where only individual red blood cells pass through[46]. Overall, we found that ADS-Vis-OCT was suitable for all vessels with a diameter greater than 35 μm as repeatability did not significantly vary between larger vessels of diameter greater than 100 μm and smaller vessels.

In Fig. 3, we show that SCs impose biases on vis-OCT STFT spectra. The SCs vary with different vessel locations, sizes, and types (e.g., artery or vein). Normalizing by local, highly reflective tissues like the anterior vessel wall (AW) does not provide repeatable biases. First, most vessels are buried under several layers of retinal tissues, like the NFL, with varying optical properties[18,47,48]. Second, interfaces at the ILM and AW are composed of highly reflective and fibrous tissues. For example, consider vessel V1 in Fig. 3. The ILM/NFL interface appears transparent at some parts of the retina but is bright directly above this vessel. Since the brightness is most intense when the interface is orthogonal to the vis-OCT illumination beam, it is likely from specular reflection or high backscattering from the fibrous tissues. A similar trend is seen at the AW interface for V1. While most of the vessel wall has similar intensity to the NFL, the center of the AW exhibits a higher reflectance. Again, this may be a specular reflection or strong backscattering from the fibrous vessel lamina[49,50]. Meanwhile, V3 in Fig. 3 demonstrates no such bright reflectances. Such light-tissue interactions can have spectral profiles dependent upon the incident angle of light, as well as local optical properties such as polarization[48,50,51]. Normalization by the signal at $z_d$ (start of blood signal decay) isolates the blood signal from the optical properties of the vessel wall and overlaying retinal tissue, which varies between different individuals. In a Monte Carlo simulation of photon propagation in blood, we demonstrated that light propagating through blood collected by vis-OCT travels the same optical path distance in the vessel wall and retina regardless of the depth traveled in the blood[34]. Thus, normalization at $z_d$ minimizes the variability of these overlaying layers between different individuals. We provide an additional example of the unpredictability of retinal backscattering in Fig. S5.

We note that removing SCs differs from the calibration procedure used in intensity-based oximetry, such as fundus photography-based oximetry. Because intensity-based oximetry is not depth-resolved, the optical properties of the retina outside the blood vessel contaminate the spectral signature of blood and need to be calibrated against[52–54]. However, the optical properties of retinal layers vary between individuals, meaning the calibration varies between individuals. The depth selection step removes (steps 6 and 7 of ADS-vis-OCT) contaminant signals outside the blood region. At the chosen depth, additional SCs, such as LCA and SDBG, are removed by evaluating the measured spectrum against physical models for these SCs. As these SCs vary between vessels, imaging conditions, and individuals, these SCs are determined adaptively in this work rather than using a rigid calibration.

Table 1 shows that the current strategies for vis-OCT retinal oximetry fail to achieve desired accuracy and repeatability in sO$_2$ measurements in our human dataset. Although the FA method attempted to remove SCs, it performed the removal without adapting measurement variables to different vessels and images. Meanwhile, the PW method indirectly measured blood's attenuation

spectrum and did not consider SCs other than the PW itself. Unfortunately, the lack of PW visibility in larger vessels (e.g., >100 μm) prevents this method from measuring sO$_2$ near the optic disk, a potential limitation to measuring total oxygen delivery and collection in the retina. Additionally, the PW method yielded significantly higher sO$_2$ uncertainty than ADS-vis-OCT but had comparable spectral fit $R^2$, meaning it overfitted the measured spectrum. Overfitting causes not only systemic sO$_2$ bias and uncertainty but also false confidence ($R^2$) in the reliability of the results. We suggest a few potential sources of such overfitting below.

First, vis-OCT measures a scattering coefficient of blood lower than that predicted by the Mie theory[22,55]. We can modify the fitted scattering coefficient by multiplying it with the factor SSF. However, the value of SSF in vis-OCT varies significantly among different studies[16,19,27,56]. Recently, we determined using Monte Carlo simulation and experimental validation that our vis-OCT system measures an SSF near 0.06[55]. However, the PW method uses SSF = 0.2, based on a model dependent only on hematocrit[16]. We believe the prior conclusion of SSF = 0.2 was the result of incomplete removal of SCs, since higher SSF results in STFT spectral correlation (see Fig. 3 in ref. [16]) with those measured using Method 1 in Fig. 3. Method 1 generally produces a negative bias at shorter wavelengths and a positive bias at longer wavelengths. When the PW method selects SSF = 0.2, it assumes that these biases are all oxygenation-dependent, which may be untrue. Any incorrect assumption of SSF may interact in the fitting model with the power law ($-\alpha\ln(\lambda)$), where $\alpha$ is an arbitrary constant), which exists to correct for the backscattering spectrum from the PW. The power law's monotonic decrease with wavelength is opposite to the bias trend seen in normalization Method 1 and the increased SSF. Therefore, if the predetermined SSF = 0.2 is overestimated, as we predict here, the power law will be overfitted to compensate for this overestimation. In this sense, SSF = 0.2 and the power law both fit common bias trends but incorrectly describe the physics of the SDA-line (see Eqs. 1 and 2), leading, in part, to inability to differentiate oxygen-dependent and non-oxygen-dependent variables in the model. We note that the power law is also susceptible to overfitting other SCs such as SDR and SDBG, which typically increase and decrease with wavelength, respectively.

Although the PW model is sensitive to sO$_2$ values in rodents[5,16,21], where SNRs are high, imaging times are extended, and vessels are homogeneous, failure to properly model the light-tissue interaction or remove SCs prevents researchers from predictably assessing sO$_2$ error or uncertainty in the clinic. Furthermore, such errors and uncertainties may vary with different vis-OCT systems or imaging protocols, challenging reproducibility.

Using ADS-vis-OCT in imaging well-controlled ex vivo blood phantoms (Fig. S1), we found spectra consistent with SSF = 0.02–0.10, nearly 10-fold less than the reported SSF = 0.2. The average fitted SSF in the phantom experiment was 0.068 at physiological hematocrit (45%) (Fig. S2a). Both of these measurements are consistent with our recent investigation of the scattering coefficient of whole blood in vis-OCT[55]. Since our normalization protocol explicitly isolated the scattering and absorption coefficients of blood from SCs, we anticipated that spectral measurements in the human retina should be comparable to the ex vivo phantoms (neglecting any spectral differences between human and bovine blood). We measured the average best-fit SSF in the human retina as 0.064 (Fig. S2), very close to that value in the ex vivo experiments (0.068) and predicted by our simulations[55]. Even though the two experimental conditions differed, we reached nearly identical quantitative conclusions, validating that ADS-vis-OCT removes SCs and appropriately models the measured data.

This work targets key sources of SCs in vis-OCT. Some SCs, as the ones that originated in the retinal tissues, are unavoidable and must be corrected for during image processing. Others, like LCA and SDR, are

hardware-specific and may have a different level of influence on sO$_2$ depending on the optical design of the system. Our system used a 0.05 numerical aperture (NA)[34], which improved lateral resolution at the trade-off of increased LCA. An alternative approach is to reduce the NA and minimize the LCA correction neccessary for sO$_2$ measurement. However, this approach may reduce the visualization of small vessels and anatomical structures of interest in the clinic. Thus, an investigation of the optimal relation between lateral resolution, aberrations, and sO$_2$ accuracy is warranted.

In summary, we took full advantage of the high-resolution and high contrast in vis-OCT to achieve adaptive, depth-resolved analysis of spectral signatures from light-blood interactions. We developed and validated ADS-vis-OCT for retinal oximetry in 18 healthy volunteers in retinal vessels ranging from 37 μm to 174 μm in diameter in a clinical environment. We found excellent spectral fits, repeatability, and agreements with the pulse oximeter readings, setting the stage for future clinical vis-OCT retinal oximetry applications.

## Data availability
Data for the main figures are included in Supplementary Data 1. Data underlying the results may be obtained from the authors upon reasonable request.

## Code availability
Custom code used in this paper may be obtained from the authors upon reasonable request.

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

## Acknowledgements

The authors thank Prof. Robert Linsenmeier for fruitful discussions. The authors are grateful for the generous support from the National Institutes of Health R01EY026078, R01EY029121, R01EY019949, R44EY026466, and U01EY033001.

## Author contributions

I.R. and H.F.Z. designed the study. I.R. and R.K. developed the ADS-vis-OCT method. I.R., R.K., R.F., Z.G., L.B., D.M., G.W., H.I., and J.S. collected and analyzed the data used in the study. I.R., R.K., R.F., and H.F.Z. revised and edited the manuscript. I.R., R.K., and Y.W. designed the software used for the study. All authors reviewed and approved the manuscript.

## Competing interests

R.V.K., Y.W., J.S.S., and H.F.Z. have financial interests in Opticent Inc. All other authors have no competing interests to declare.
