## [Peer Review File · Communications Medicine]

Reviewers' comments:

Reviewer #1 (Remarks to the Author):

Dear authors

I find your manuscript very interesting and well written. I find no major flaws, although I must admit that I am not an expert on the technical details.

1. The method gives repeatable and credible results. It would have been very interesting to see the results of inhalation of pure oxygen or some other stimulus that could have more clearly demonstrated sensitivity of the method. If the authors have some data on sensitivity, could that be included?
2. The lower saturation in smaller retinal arterioles is interesting. I agree that direct diffusion of oxygen from arterioles can have an effect on saturation in general. However, from figure 4 it seems that there is a rather abrupt change in saturation from for example vessel 5 to 6 or from 9 to 10. One would assume that if you measure saturation on both sides of a vessel branching (close to the branching) the saturation should be rather similar in all three branches. Is this the case in general in your data?

Reviewer #2 (Remarks to the Author):

The authors have conducted a comprehensive pre-clinical test of visual-spectrum OCT oximetry. The adaptive analysis approach is described in detail and the results are clearly presented.

1. It appears that the individual vessel segments are treated as independent observations in the analyses. Could it be appropriate to see whether measurements differed between individuals?
2. Although oxygen saturations above 100% are not biologically feasible, the authors could consider whether exclusion of such measurements is the best solution. In dual-wavelength photographic oximetry such oxygen saturations are occasionally seen. Typically, in these cases, venous oxygen saturations are generally higher as well. Therefore, the arterio-venous oxygen saturation difference is often presented as a more robust measure. It is important to stress that the saturation values which are achieved with retinal oximetry are estimates based on mathematical modelling of measured light reflectivity.
3. The repeatability noted in table S2 appears to be based on a relatively small number of vessels. Would it be possible to increase the number of vessels to ensure that selection of vessel segments does not influence the results?
4. How accurate are the measurements from vessels that are overlying each other?
5. Is it possible that the SO₂ in retinal arteries could be lower than the SO₂ measured with a

pulse oximeter? Could there be individual differences?

6. Would it be relevant to introduce a vessel diameter correction in the vis-OCT SO₂ calculation or to exclude vessels below a certain diameter?

Reviewer #3 (Remarks to the Author):

See attached file

Review of Adaptive spectroscopic visible-light optical coherence tomography for clinical retinal oximetry

The manuscript describes work towards improving the accuracy and repeatability of in vivo retinal vascular oximetry using OCT, in particular addressing the issues of so-called spectral contaminants. The primary audience would appear to be researchers developing new measurement techniques (typically physicists and engineers) and will not be very accessible to those without these skills. The emphasis of the manuscript is not on demonstrating the medical benefit of the improved oximetry.

Overall I consider the paper to report interesting advances in the field OCT oximetry : the issue of extracting reliable signatures from images within the retina is quite challenging, requiring many complex interactions to be addressed. The article addresses and mitigates many of the main influences, but also omits some issues without comment.

It is stated in the introduction that there is a requirement for more accurate oximetry since this provides a potential for detection of several retinal diseases before irreversible vision loss occurs. This is not an uncommon claim of retinal oximetry, but is a little difficult to justify – for example, the cited reference 9 refers to glaucoma for which ~10% variation in venous oximetry corresponds to the progression from perfect health to near blindness - however there may be inter-subject and intra-subject variability in oximetry that exceeds the measurement precision and accuracy of about 3% (as demonstrated in this paper by Fig 4B) - I do not believe that there is any evidence that this provides early warning for glaucoma compared to eg measurement of RNFL or a visual field test. This begs the question what improved oximetry, the main aim of this paper, would provide compared to traditional techniques. This does not detract from the quality of the work, but given the target audience of the journal, the manuscript would benefit from more care with the claims.

A key contribution is mitigation of so-called spectral contaminants, which are separated into systemic and eye effects. This seems a strange terminology: in particular systemic “spectral contaminants” seems to be effectively accounting for the system spectral transfer function and eye-interface transfer functions. (i.e. a multiplicative effect) through a normalization process. The main body of the manuscript does not demonstrate sufficiently how its aims differ from ‘calibration’ of system and eye effects. The manuscript would benefit (within the main text) from (1) a clearer explanation of the physics behind these spectral contaminants and importantly, (2) setting the context of the benefits of OCT-based oximetry compared to the traditional SLO or fundus-camera-based oximetry, which is significantly more mature. In particular, the following articles have addressed the issues of the spectral effects of the complex structure within the eye (and also their calibration and normalization): Beach et al *Journal of Applied Physiology* **86**, 748-758 (1999), Smith, M. H et al *Appl Optics* **39**, 1183-1193 (2000), Carles, Get al. *J Biomed Opt* **22**, 116002 (2017). The issues are different from OCT oximetry, but seem to have achieved at least the same level of performance. One would expect that compared to simple intensity-based oximetry, OCT offers potential for a substantial improvement for accuracy, and repeatability due to the ability to gate out contributions from multiple complex light paths and it would enhance the manuscript to recognise the two different approaches. In particular, the stated accuracies and repeatabilities seem to be comparable to measurements using established hyperspectral, multispectral and two-wavelength oximetry – in fact the repeatabilities seem a little worse.

The section describing the algorithm provides welcome detail, but the absence of the overall context and the bigger picture makes this unengaging and would be more suited as an appendix. Understanding would be enhanced by a top-level discussion of the principles.

For the results described in Fig 3A, it seems that conclusions have been reached on a very small data set. There is immense variability in optical clutter and associated fitting both within a single eye, between nominally similar healthy eyes and between eyes that vary with health, ethnicity and age. It is important to account for such variabilities in forming reliable empirical conclusions and this is not adequately addressed here, although the sample size may be reduced with the benefit of a rigorous description of the physics of light propagation in the retina.

The ability to conduct distinct oximetries in vessels 3 and 4 in Fig 4 is interesting, but the oximetry of 85.8% seems to be much lower than systemic oxygenation seems improbable and merits more rigorous. The hypothesis that lower SO_2 is due to diffusion is not substantiated and could also be due to short comings in measurement accuracy, which is an important claim of the paper. Evidence in support of the proposed hypothesis could be an oxygenation gradient along the length of the vessel – which would be expected from diffusion, since the arterial blood at the exit of the ONH should be close to saturated. Other researchers (eg Einar Steffanson's group) have hypothesised oxygen counter currents along the length of arteries due to oxygen diffusion to veins. Laminar flow also results in variations in oximetry across the widths of some venules (due to different venules at branches draining different parts of the retina with dissimilar oxygen consumptions) - this is not evident in Fig 4A, but would lend support to the robustness of the technique.

What was the diversity of the subjects (age, ethnicity, etc.)?

The paper describes some inconsistencies between measurements of blood optical properties. These differ for polarised and unpolarised light – how is polarisation incorporated into the discussions? Most Monte Carlo modelling does not include polarisation and in the ballistic propagation regime relevant to OCT there are significant differences between unpolarised and polarised light propagation. What is the justifications for attributing differences in SSF to “spectral contaminants” rather than the other imperfections in the physical model (see also comments below on RBC alignment). How does the polarisation state within the ex vivo measurements compare to polarisation for OCT in the eye (which is generally less well controlled due to birefringence effects). It is striking that 100% OS values all have SD of 0% - is this because these measurements were artificially limited to 100% If so, I think this should be avoided – it is not uncommon for noise to result in oximetries >100%. Indeed the oxygen dissociation curve prevents OS achieving 100% (>99.95%) so 100% is also non physical, just as is >100%.

Back scatter from RBCs is also effected by laminar flow causing the alignment of RBCs parallel to the vessel walls causing a characteristic figure-eight B-scan reflectivity pattern within the blood vessels – particularly arteries. How is this accounted for within the model?

The paper emphasises the importance of measuring total oxygen delivery to the retina by oximetry of the vessels close to the optic nerve, but omits to mention the fact that total delivery requires also a measurement, or inference, of flow and also that a substantial, but variable (auto-regulated) oxygen supply to the retina is provided by the choroid.

How were the measurements in vessel segments aggregated into a single measurement – was it a single mean or was there a process to remove outliers?

Minor issues

1. “ $\sqrt{I_{samp}(\lambda)}$ and $\sqrt{I_{ref}(\lambda)}$ are the power spectra of the light collected from the sample and reference arms, respectively.” – doesn't the square root make these amplitude spectra?
2. “The term $-\mu_{t_{blood}}(\lambda) \left[(z_0 - z_d) + \frac{\Delta z}{2} \right]$ is a constant and represents uniform, oxygen-dependent attenuation along the vessel depth. Blood cell packing, orientation, flow, and oxygen diffusion may add variability to this assumption. “ It could be highlighted that due to the aforementioned laminar flow, oxygenation may vary substantially within venules.
3. “consistent with our previous findings ()” – missing reference

Reviewer #1

Dear authors, I find your manuscript very interesting and well written. I find no major flaws, although I must admit that I am not an expert on the technical details.

Reply: We thank this reviewer for the appreciation of our work.

1. *The method gives repeatable and credible results. It would have been very interesting to see the results of inhalation of pure oxygen or some other stimulus that could have more clearly demonstrated sensitivity of the method. If the authors have some data on sensitivity, could that be included?*

Reply: We agree with the reviewer that external stimulation-based imaging studies in human eyes are critical to investigate the retinal hemoglobin oxygenation (sO₂) imaging sensitivity. We plan to conduct oxygen challenging and light stimulation studies in volunteers as the next stage after overcoming the stability obstacle. This particular work focused on solving the repeatability issue in human sO₂ imaging induced by the newly-discovered spectral contaminants. As we classify the spectral contaminants into imaging system-dependent and sample-dependent contaminants, we found that direct calibration or normalization is insufficient to remove the influence of spectral contaminants, especially the sample-dependent ones. Since the sample-dependent spectral contaminants vary with different eyes, we found that our iterative method best overcame the sample-dependent challenges through stepwise optimization. However, we have tested the sensitivity of our sO₂ measurement in well-controlled *ex vivo* blood samples against the gold stand blood gas machine. In our *ex vivo* phantom study, the regression slope was 1.01 ± 0.024 , leading to the sensitivity measurement in phantom of 2.4%. We expect sensitivity in humans to be similar although further verification is needed. This is now reported in the "*Ex vivo* phantom validation and *in vivo* comparison" supplemental of the revised manuscript.

2. *The lower saturation in smaller retinal arterioles is interesting. I agree that direct diffusion of oxygen from arterioles can have an effect on saturation in general. However, from figure 4 it seems that there is a rather abrupt change in saturation from for example vessels 5 to 6 or from 9 to 10. One would assume that if you measure saturation on both sides of a vessel branching (close to the branching) the saturation should be rather similar in all three branches. Is this the case in general in your data?*

Reply: We thank the reviewer for pointing out the abrupt color changes between vessels 5 and 6 and vessels 9 and 10 in the pseudo-colored Figure 4. We double-checked our results and found that the measured sO₂ values on both sides of the vessel branching are comparable under the expected SNR influence. The reason why the colors appear to have abrupt changes is because we assigned the color of each vessel segment using the mean values of the measured sO₂ within the entire corresponding vessel segments. Due to intrinsic variations in the numerical inverse calculation of sO₂ from vis-OCT spectral data, we averaged the sO₂ values within the entire vessel segment. In future work, we plan to explore the optimal vessel segment length for sO₂ averaging and will develop better strategies to perform spatial averaging around vessel branching points.

Reviewer #2:

The authors have conducted a comprehensive pre-clinical test of visual-spectrum OCT oximetry. The adaptive analysis approach is described in detail and the results are clearly presented.

Reply: We thank this reviewer for the appreciation of our work.

1. *It appears that the individual vessel segments are treated as independent observations in the analyses. Could it be appropriate to see whether measurements differed between individuals?*

Reply: We appreciate the suggestion from the reviewer. In this work, we treated each vessel segment independently. We particularly focused on identifying and removing spectral contaminants in human vis-OCT images and developed the reported iterative algorithm to achieve this goal. As shown in Fig. 5, we summarized measurements from all human subjects and examined the trend of sO₂ changes with vessel diameter, and compared the sO₂ values in major arteries, small arteries, and veins. As a result, we hypothesize that our new method is stable among different subjects. In Table R1, we show mean sO₂ values in major arteries, small arteries, and veins for each subject. For individuals, the mean sO₂ in major arteries was 97.4 ± 0.8 , in small arteries was 92.4 ± 1.4 , and in veins was 57.6 ± 0.6 . We found that the sO₂ values in major arteries remain comparable among all human subjects. The sO₂ values in small arteries and veins showed greater variation among the same human subjects. We are currently collecting data to obtain baseline and variations of retinal sO₂ values in humans with different ages, races, and glaucoma stages. We will report the new results as soon as they are ready.

Subject	Major artery (mean \pm Standard error)	Small artery (mean \pm Standard error)	Vein (mean \pm Standard error)
1	99.4 ± 0.5	84	56.8 ± 1.2
2	99.7 ± 0.3	NA	56.7 ± 2.5
3	98 ± 1.2	94.3 ± 2.1	57.4 ± 2
4	95.2	NA	60.1 ± 8.2
5	98.3 ± 1.3	100	60.1 ± 0.5
6	98.6 ± 1.4	NA	58.4 ± 2.4
7	100 ± 0	NA	57 ± 1.5
8	96.5 ± 1.5	NA	60.2 ± 1.7
9	98 ± 2	92.1 ± 0.9	62 ± 2.7
10	96.1 ± 1	NA	58.5 ± 2.1
11	NA	91.4 ± 3.6	53.5 ± 1.1
12	98 ± 2.1	100	54.1 ± 1.1
13	98.9 ± 0.5	86.4	58.4 ± 3
14	97.8 ± 0.9	93.1 ± 1.3	59.3 ± 1
15	NA	96 ± 2.5	60 ± 2.6
16	86.7	91.1 ± 1.7	51 ± 1.3
17	99.3 ± 0.7	92.3 ± 2.0	58.7 ± 1.3
18	97.4	87.5 ± 2.5	54.2

Table R1. Comparison of mean sO₂ values in major arteries, minor arteries, and veins among different human subjects.

2. *Although oxygen saturations above 100% are not biologically feasible, the authors could consider whether exclusion of such measurements is the best solution. In dual-wavelength photographic oximetry such oxygen saturations are occasionally seen. Typically, in these cases, venous oxygen saturations are generally higher as well. Therefore, the arterio-venous oxygen saturation difference is often presented as a more robust measure. It is important to stress that the saturation values which are achieved with retinal oximetry are estimates based on mathematical modelling of measured light reflectivity.*

Reply: We agree with the reviewer that oxygen saturation above 100% is not feasible. In our algorithm, we did not ever measure sO_2 above 100%. The maximum value is $sO_2 = 100\%$ because it is the nature of the non-negative least-squares regression algorithm to bound sO_2 between 0% and 100% (V. Esposito Vinzi, G. Russolillo, Partial least squares algorithms and methods, *WIREs Computational Statistics* **5**, 1-19, 2013). We chose a non-negative least-squares algorithm with the aim to be automatically within the physiological range of the sO_2 values without any additional manipulations. Although $sO_2 = 100\%$ is technically non-physical compared with the physical limit of 99.95%, we, nor any other technology has the precision to differentiate between the two and it is therefore negligible. In the manuscript, we recognized the potential statistical influences of the upper bound $sO_2 = 100\%$ and actively increased the strictness of our quality control when $sO_2 = 100\%$ to minimize the effect of an $SD = 0\%$ (see Supplementary information: *Adaptive Filtering, Stage 2*). Moreover, we compared sO_2 values in major arteries to pulse oximeter readings (see Supplementary Information: Table S2), where higher measured sO_2 (e.g. $> 98\%$) typically coincided with such high values from the pulse oximeter. To clarify this, we added an additional statement in the "Oximetry fitting model" section.

3. *The repeatability noted in table S2 appears to be based on a relatively small number of vessels. Would it be possible to increase the number of vessels to ensure that selection of vessel segments does not influence the results?*

Reply: We agree with the reviewer that increasing the number of independent vessels will be helpful. In the reported study, we imaged 125 individual vessels from 18 subjects. Among the 125 individual vessels, we validated 42 vessels with a pulse oximeter attached to 13 subjects during data acquisition. In addition, we performed a large amount of blood phantom imaging and validated the vis-OCT sO_2 measurement by the gold-standard blood-gas analyzer. In the phantom studies, we acquired 100 measurements at each of the 14 different sO_2 levels. Based on these data, we have found good agreement of the scattering scaling factor between blood phantom measurements and human measurements, which cannot be reached using a non-iterated method as described in the manuscript. Since we found that spectral containments contributed the most to the repeatability, finding agreement between phantom and human measurements in the scattering scaling factor represents a major step forward. As a result, our iterative method produced highly repeatable sO_2 measurements ($\leq 2.5\%$) across a broad range of vessel types and agreed within 2.1% of pulse oximeter readings in major arteries, which shows that the numbers of human subjects and independent vessels are sufficient for this purpose. Currently, we continue to acquire human data and will report the sensitivity of our new sO_2 measurement when ready.

4. *How accurate are the measurements from vessels that are overlying each other?*

Reply: Based on the vessel sizes of vessel 3 and vessel 4, their respective sO₂ measurements (98.3% and 85.8%) are comparable to other vessels with similar diameters. However, there has been no gold standard to validate retinal sO₂ in humans. We are confident that our new iterative algorithm successfully differentiated these two vessels, although they spatially overlay horizontally. The depth resolution of vis-OCT renders measurements from both these vessels independent observations, meaning the accuracy of these overlaying vessels is similar to non-overlapping vessels. The depth resolved nature of the measurements is noted in the "Retinal oximetry around the optic disk" section, which states "We measured sO₂ values from both vessels, demonstrating the unique depth-resolved sO₂ imaging capability permitted by vis-OCT." This observation confirmed the effectiveness of the iterative processive to reject sample-dependent spectral contaminants.

5. Is it possible that the SO₂ in retinal arteries could be lower than the SO₂ measured with a pulse oximeter? Could there be individual differences?

Reply: We conducted a thorough review of the literature reporting comparing sO₂ values in retinal arteries with pulse oximetry readings. We found that the values are reported to be comparable, although sO₂ in retinal arteries is believed to be possibly slightly lower than pulse oximetry readings. Traustason et al. found that mean retinal arterial readings using a fundus camera-based retinal oximetry were 5% lower than the readings from the pulse oximetry attached to fingers. However, the authors did not find the difference to be statistically significant (Traustason, et al. "Retinal oxygen saturation in patients with systemic hypoxemia." IOVS 52.8, 5064-5067, 2011). Another study comparing retinal oximetry with invasive femoral artery measurements found that retinal oximetry measurements were 2% lower, but the difference was also not statistically significant (Eliasdottir et al. "Retinal oximetry measures systemic hypoxia in central nervous system vessels in chronic obstructive pulmonary disease." PLoS One 12.3, e0174026, 2017). Slightly lower retinal oximetry values that were not statically significant have been reported by other authors in chronic obstructive pulmonary disease (COPD) patients with and without supplemental oxygen (Eliasdottir. "Retinal oximetry and systemic arterial oxygen levels." Acta Ophthalmologica 96, 1-44, 2018). In our study, major arteries had slightly lower oxygenation levels (98.3%) than the pulse oximetry readings (98.6%). Factoring in the fact that smaller arteries had an average sO₂ of 93.2%, our method shows similar consistency between pulse oximetry and retinal oximetry. While there could be individual differences between pulse oximeter readings and retinal sO₂ values, no current technology has the accuracy and precision to evaluate this reported difference between retinal oximetry and pulse oximetry since the mean error associated with pulse oximetry compiled across multiple studies exceeds 3% (Nitzan, et al, "The various oximetric techniques used for the evaluation of blood oxygenation," Sensors 20.17, 4844, 2020). Given that pulse oximetry and retinal oximetry values are comparable in COPD patients in both normoxic and hyperoxic breathing, it is likely that any difference between individuals is within the precision associated with measurements of sO₂ (Eliasdottir, et al., "Retinal oximetry and systemic arterial oxygen levels," Acta Ophthalmologica 96, 1-44, 2018).

We added additional discussion to the discussion section to clarify the difference between retinal sO₂ and pulse oximetry in the revised manuscript.

6. Would it be relevant to introduce a vessel diameter correction in the vis-OCT SO₂ calculation or to exclude vessels below a certain diameter?

Reply: We excluded vessel diameters below 35 micrometers in our sO₂ measurement. For vessels less than 35 micrometers, we found that the backscattered optical spectra became unstable, likely induced by orientation-dependent backscattering from rotating red blood cells. This effect is amplified at the capillary level where only individual red blood cells can pass through. (Liu et al., "Theoretical model for optical oximetry at the capillary level: exploring hemoglobin oxygen saturation through backscattering of single red blood cells." *J Biomed Opt* 22, 25002, 2017.) To inversely calculate blood oxygenation from the spectra of oxygenated and deoxygenated hemoglobin, blood is assumed to be a homogeneous and continuous media. We found that when the vessel diameter is less than 35 μm , we cannot consider blood to be a continuous medium. For vessels larger than 35 μm , we found our iterative algorithm performed equally well in nearly all vessel diameters. We added more discussion about vessel size in the second paragraph of the discussion section to the revised manuscript.

Reviewer # 3

Review of Adaptive spectroscopic visible-light optical coherence tomography for clinical retinal oximetry

The manuscript describes work towards improving the accuracy and repeatability of in vivo retinal vascular oximetry using OCT, in particular addressing the issues of so-called spectral contaminants. The primary audience would appear to be researchers developing new measurement techniques (typically physicists and engineers) and will not be very accessible to those without these skills. The emphasis of the manuscript is not on demonstrating the medical benefit of the improved oximetry.

Reply: We thank this reviewer for the appreciation of our work.

1. *Overall I consider the paper to report interesting advances in the field OCT oximetry: the issue of extracting reliable signatures from images within the retina is quite challenging, requiring many complex interactions to be addressed. The article addresses and mitigates many of the main influences, but also omits some issues without comment.*

It is stated in the introduction that there is a requirement for more accurate oximetry since this provides a potential for detection of several retinal diseases before irreversible vision loss occurs. This is not an uncommon claim of retinal oximetry, but is a little difficult to justify – for example, the cited reference 9 refers to glaucoma for which ~10% variation in venous oximetry corresponds to the progression from perfect health to near blindness - however there may be inter-subject and intra-subject variability in oximetry that exceeds the measurement precision and accuracy of about 3% (as demonstrated in this paper by Fig 4B) - I do not believe that there is any evidence that this provides early warning for glaucoma compared to eg measurement of RNFL or a visual field test. This begs the question what improved oximetry, the main aim of this paper, would provide compared to traditional techniques. This does not detract from the quality of the work, but given the target audience of the journal, the manuscript would benefit from more care with the claims.

Reply: We agree with the reviewer that the impact of retinal oximetry on glaucoma management remains understudied due to the lack of reliable clinical tools. Evidence that oximetry improves

glaucoma management depends on robust measurements that do not yet exist, and this work aims to provide a method to obtain such measurements. However, as glaucoma progresses, patients lose retinal ganglion cells, leading to detectable reduced retinal nerve fiber layer thickness. Therefore, an educated guess is that the inner retinal oxygen demand will drop correspondingly, and improved retinal oximetry has the potential to detect such oxygen demand alteration. Our next step is to investigate the accuracy of oximetry *in vivo* as well as to optimize the imaging protocol required to provide clinically valuable information for different retinal diseases with impaired oxygen delivery and consumption.

Retinal oximetry will supplement existing technologies rather than replace them. Diagnosis and management of diseases are based on multiple observations (e.g., intraocular pressure, retinal nerve fiber layer thickness, ganglion cell layer thickness, visual field testing, central corneal thickness for glaucoma). As long as oximetry helps to improve the outcome of managing glaucoma and other retinal disorders in conjunction with existing measurements, it has clinical relevance worth of exploring. Moreover, reliable and accurate oximetry (level of sensitivity needs to be established in future studies) will potentially assist in the diagnosis and management of other oxygen-dependent diseases, including diabetic retinopathy (Duh *et al.*, "Diabetic retinopathy: current understanding, mechanisms, and treatment strategies," *JCI Insight*, e93751, 2017), retinopathy of prematurity (Hartnett *et al.*, "Effects of oxygen on the development and severity of retinopathy of prematurity," *J AAPOS*, 229-34, 2013), and age-related macular degeneration (Stefánsson *et al.* "Metabolic physiology in age related macular degeneration," *Prog Retin Eye Res*, 72-80, 2011) among others.

*2. A key contribution is mitigation of so-called spectral contaminants, which are separated into systemic and eye effects. This seems a strange terminology: in particular systemic "spectral contaminants" seems to be effectively accounting for the system spectral transfer function and eye interface transfer functions. (i.e. a multiplicative effect) through a normalization process. The main body of the manuscript does not demonstrate sufficiently how its aims differ from 'calibration' of system and eye effects. The manuscript would benefit (within the main text) from (1) a clearer explanation of the physics behind these spectral contaminants and importantly, (2) setting the context of the benefits of OCT-based oximetry compared to the traditional SLO or fundus-camera based oximetry, which is significantly more mature. In particular, the following articles have addressed the issues of the spectral effects of the complex structure within the eye (and also their calibration and normalization): Beach *et al* *Journal of Applied Physiology* **86**, 748-758 (1999), Smith, M. H *et al* *Appl Optics* **39**, 1183-1193 (2000), Carles, *Get al.* *J Biomed Opt* **22**, 116002 (2017). The issues are different from OCT oximetry, but seem to have achieved at least the same level of performance. One would expect that compared to simple intensity-based oximetry, OCT offers potential for a substantial improvement for accuracy, and repeatability due to the ability to gate out contributions from multiple complex light paths and it would enhance the manuscript to recognise the two different approaches. In particular, the stated accuracies and repeatabilities seem to be comparable to measurements using established hyperspectral, multispectral and two-wavelength oximetry – in fact the repeatabilities seem a little worse.*

Reply: We thank the reviewer for highlighting the influence of the spectral contaminants (SCs), which we defined in the manuscript as "any erroneous spectra not associated with blood's optical attenuation". In fact, our manuscript specifically addresses the question of how "*the main body of the manuscript does not demonstrate sufficiently how its aims differ from 'calibration' of system*

and eye effects". We believe that 'spectral contaminants' is an appropriate term. The vis-OCT-based oxygenation sensing is based on the extracting of the spectral alterations induced by the oxygenated and deoxygenated hemoglobin on the sample arm vis-OCT light in retinal blood vessels. Any unaccounted spectral alterations induced on the propagating light from the vis-OCT source to the detectors by either the system or the eye potentially led to errors in the oxygen sensing. Therefore, such spectral modifications contaminate alterations induced by the blood vessel's hemoglobin, i.e., our signal. This manuscript aims to account for such SCs adaptively and robustly extract only the spectral alteration induced by the hemoglobin to accurately evaluate sO₂ levels in retinal blood vessels.

We classified SCs into two categories: sample-based and system-based, which, as the reviewer correctly mentioned, describe influences from the vis-OCT system and eye (including the retina), respectively. Although we agree that calibrations against the known system or eye attenuations are important, their efficacies rely on previously known or highly repeatable measurements. We found such calibrations were overly simplistic and insufficient for reliable human retinal oximetry. This is a major motivation of our manuscript and is discussed in detail in the Sections "Optimal tissue normalization" and "Comparison with non-adaptive retinal sO₂ measurements".

Two facts support the need for an adaptive method. First, human eyes significantly vary in structure and optical properties (such as aberration) leading to nontrivial spectroscopic signatures (contaminants). Second, variable imaging conditions (e.g. eye position relative to the vis-OCT sample beam and motions) alter these signals across imaging locations and acquisitions.

As reported in our original manuscript, we show that SCs vary among different vessels in the same retina (Fig. 3). This may be owed to changes in tissue reflections or attenuations, changes in optical focus, obliqueness of the vessel with respect to the scanning beam, etc. Such variable SCs cannot be calibrated against since they are unknown, are impractical to control with sufficient precision, and are different among vessels in the same eye. Previous failures to rigorously define and account for the SCs unique to each vessel fundamentally limited sO₂ accuracy and repeatability.

We demonstrated the need for an adaptive method quantitatively by directly comparing two previous calibration-based methods to our adaptive spectroscopic (ADS) approach (see Section "Comparison with non-adaptive retinal sO₂ measurements" and Table 2). The first compared method is the posterior wall (PW) method, which extracts sO₂ using reflectance from the posterior vessel wall. The second compared method is the fixed attenuation (FA) method, which extracts sO₂ by singular measurement of blood's attenuation coefficient along a predefined depth range, which is the basis of all retinal oximetry measurements. Table 2 and Fig. 3 in the original method show that PW and FA measurements are significantly biased and less repeatable compared with the ADS measurements.

As correctly noted by the reviewer, the SCs investigated in this work are depth-dependent, with each tissue layer contributing a different spectral contamination signature. Since blood is embedded within these layers, its spectral signature (signal) must be specifically isolated from the other contaminating signatures (SCs). Bearing in mind that SCs spectral alternations are greater than a few percentiles as compared with signal spectral alternation, such isolation can only be done using technology with sufficient depth resolution and depth penetration, such as OCT.

When retinal oximetry is performed by a two-dimensional (2D) imaging technology, such as SLO or fundus photography (PF), the SCs cannot be decoupled from the blood signal. In Fig. 1 we highlighted potential reflectance/attenuations from the inner-limiting membrane (ILM), nerve

fiber layer (NFL), ganglion cell layer (GCL), nuclear layers (NL), photoreceptor layers (PRL), and retinal pigment epithelium/Bruch's membrane (RPE/BM). Since vis-OCT is depth-resolved at 1.3- μm axial resolution, it experiences SCs only from tissues in or above the vessel and excludes the PRL, RPE, and BM. Nevertheless, in vis-OCT, we show that sO_2 measurements are unreliable when influenced by SCs in or above the vessel (please refer to the Comparison with non-adaptive retinal sO_2 measurements section).

In SLO- and PF-based retinal oximetry, however, the signal is further affected by the photoreceptor layer, retinal pigment epithelium, Bruch's membrane, and choroid. These anatomical layers are some of the most reflective and absorptive layers in the retina, as can be confirmed by all those layers resolved vis-OCT imaging of the retina (Rubinoff *et al.*, "Speckle reduction in visible-light optical coherence tomography using scan modulation Neurophotonics," 6, 041107, 2019). Thus, SCs have a stronger influence on SLO and PF than on a depth-resolved imaging modality, such as vis-OCT (Liu *et al.*, "Accuracy of retinal oximetry: a Monte Carlo investigation," Journal of Biomedical Optic 18, 066003, 2013.) Although, as the reviewer points out, researchers calibrated against some of these contaminants in 2D retinal oximetry, these calibrations are not equivalent or as rigorous as those presented in this work. More importantly, SLO- and PF-based retinal oximetry has not been tested and validated in well-controlled blood phantom studies against the gold-standard blood gas analyzer as presented in this study.

We highlight key differences with the articles cited by the reviewer below. Beach *et al.*, Journal of Applied Physiology 86, 748-758, 1999 used dual-wavelength PF to measure oxygen-dependent signals in the retina. This paper recognizes potential contamination by RPE pigmentation and normalized (divided) by signal outside the blood vessel. The authors demonstrated less variance with systemic sO_2 after normalization. However, this method assumes that all vessels in the retina have the same SCs, which we show in Fig. 3 to be incorrect. Furthermore, normalizing by such a signal is equivalent to dividing by the optical density (OD) of all tissues outside the blood vessel. It does not account for the backscattering/reflectance properties of any tissues in the vessel, including vessel walls, cell-free zone as well as non-exponential light attenuation within the blood at the edges which can be angle and tissue-dependent (Fang *et al.*, "Multiple forward scattering reduces the measured scattering coefficient of whole blood in visible-light optical coherence tomography," Biomed. Opt. Express 13, 4510-4527, 2022). Furthermore, the normalizing signal includes OD information from retinal capillary layers and the vessel-dense choroid, which contain spectral alterations induced within the blood vessels that are outside of the region of interest (ROI). In this case, local oxygen changes in the capillary or choroidal layers may be unintentionally attributed to the measured vessel oxygenation (Liu *et al.*, "Accuracy of retinal oximetry: a Monte Carlo investigation," Journal of Biomedical Optic 18, 066003, 2013.)

Addressing comparable sensitivity measurements between vis-OCT and PF, we note that Beach *et al.* calculated sensitivity from the combined slope of ODR to sO_2 calibration curve of their seven subjects in Figure 8b, where ODR is the optical density ratio and sO_2 is the ground truth oxygenation. The calibration curve acquired by Beach *et al.* was $\text{ODR} = -(0.00504 \pm 0.00029) * \text{sO}_2 + 0.617$. From this equation, the standard error of the sO_2 is 5.8% of the fitted value, equating to a sensitivity of 5.8% for the oxygenation measurements. In our *ex vivo* phantom study, we found $\text{sO}_{2,\text{ADS-vis-OCT}} = (1.01 \pm 0.024) * \text{sO}_2 + 1.28$, where $\text{sO}_{2,\text{ADS-vis-OCT}}$ is the sO_2 obtained by our method (see "Ex vivo phantom validation and in vivo comparison" in supplemental info). The standard error for the fitted slope is 2.4% of the fitted value, leading to the sensitivity measurement in the phantom of 2.4% for our method. Using this comparison, we found that our sensitivity exceeds fundus-camera-based oximetry by 2.5-fold. Additionally, the repeatability of our method

was also better. In Beach *et al.*, for a corrected ODR measurement (Fig. 8, B) of 0.2, the so₂ values are almost homogeneously distributed between 77% and 100%. While in our study (Fig. 5, C), repeatability is below 2.5% for all vessel types.

Finally, we note that Beach *et al.*'s work is limited in scope compared with our study. First, Beach *et al.* measure sO₂ only in a few of the retina's largest vessels (5 arterio-venous measurements in Table 4). Second, their method assumes a calibrated relationship between OD and sO₂. This differs from our measurement of absolute sO₂ value, which is free from complex *a priori* assumptions or pre-calibrations. Since the exact relationship between OD and sO₂ in PF is unknown in each eye and each blood vessel, any errors in the calibration are propagated into the final measurement. Any unpredictability or uncertainty of the source of a systemic bias in the retinal sO₂ measurements can hinder medical decisions in a clinical environment. In contrast, our work actively isolated blood signals from all other SCs, essentially making the measurements environment-independent. Third, Beach *et al.* did not validate their finding in a well-controlled *ex vivo* experiment. Such validation is critical for confirming the validity of a retinal oximetry method since there is no golden standard for *in vivo* sO₂ measurements. We validated our results with *ex vivo* measurements in a phantom and demonstrated negligible bias and high repeatability. (see Supplementary Information: "Ex vivo phantom validation and *in vivo* comparison"). Taking the limited scope of Beach *et al.* work, its quantitative limitations, and uncertainties in systemic bias, we respectfully disagree that our "repeatabilities seem a little worse".

Smith, M. H *et al.* *Appl Optics* 39, 1183-1193, 2000 develop an analytical model to correct for complex light paths in retinal blood vessels, including 'double passes' of photons through the blood vessel. They use SLO, which provides better lateral resolution and lateral photon gating than PF; but still lacks depth resolution comparable to OCT. Owing to the limited depth resolution, authors must make assumptions about SCs as they are unable to isolate the blood signal from other tissues at the same lateral position. This contrasts with our approach of using high-resolution depth gating to filter out SCs. Smith *et al.* tested their model in an *in vitro* eye model and showed that their corrections significantly improved the accuracy of retinal oximetry measurements. However, when applying their calibrated model to an *in vivo* swine eye, the model systematically overestimated sO₂. Smith *et al.* addressed this issue by calibrating their swine model to the sO₂ measurements from the femoral artery of the same swine. However, they did not test their new calibrated swine model in additional eyes. Their data shows that the calibration from the model eye may not be applicable to all *in vivo* eyes. Further, they did not show the calibration in the swine eye resulted in accurate measurements in other swine eyes. This suggests that calibrations in the swine eye may not be transferrable to humans, and further calibration in a specific individual may not be valid in different individuals. Based on our results in Fig. 3, calibration must be done for each retinal blood vessel for each data set since intra-imaging conditions are different. Thus, applying a single (universal) analytical model for retinal oximetry, as reported by Smith *et al.*, cannot be used for robust *in vivo* measurements.

A key advantage of ADS-vis-OCT is its ability to adapt to unique vessel environments and imaging conditions. This illustrates a key distinction of how our method differs from a 'calibration,' whereas a 'calibration' fits the measured transmittance spectrum to known sO₂ measurements and is inaccurate when eyes and imaging conditions of each blood vessel differ. ADS-vis-OCT calculates the SCs based on the physics of light propagation automatically adapted for each vessel and can give accurate oximetry without the need for preliminary calibrations. Table 2 and Fig. 3 show that a rigid calibration based on mathematical modeling and normalizations alone is insufficient for accurate and repeatable *in vivo* retinal oximetry, which is achieved by ADS-vis-

OCT. Furthermore, our work, unlike the studies mentioned by the reviewer, or any other vis-OCT articles to our knowledge, is the first to cross-validate its results across four key modes: mathematical modeling, Monte Carlo simulation (Fang *et al.*, "Multiple forward scattering reduces the measured scattering coefficient of whole blood in visible-light optical coherence tomography," Biomed. Opt. Express 13, 4510-4527, 2022,) *ex vivo*, and *in vivo* measurements.

Get *et al.* J Biomed Opt 22, 116002, 2017, similar to Smith *et al.*, used SLO towards the goal of human retinal oximetry. They developed a new SLO schematic to filter out some retinal reflections and scatterings to enhance blood signals measured in vessels. Notably, they measured blood contrast *in vitro* and *in vivo* but did not measure sO₂. Get *et al.* made qualitative assessments that may help in sO₂ calculations but did not show evidence for this. Although this work provides a nice advancement of SLO image quality, we believe it is outside the scope of our work and does not demonstrate equivalent methods as claimed by the reviewer.

Finally, our statements above are supported by a recent review paper on 2D retinal oximetry (Garg *et al.*, "Advances in retinal oximetry," TVST, 10, 5, 2021). This article comprehensively reviews early retinal oximetry to the current state of the art in experiments from 1963 through 2017. Garg *et al.* found systemic disagreements between measurements from PF-based human retinal oximetry. In particular, in 2014-2017 alone, they found average sO₂ measurements in arteries ranging from 85.5% - 97.0% and average sO₂ measurements in veins ranging from 48.2 % – 60.4 %. Furthermore, they indicate that in Hammer *et al.*, "Retinal vessel oximetry-calibration, compensation for vessel diameter and fundus pigmentation, and reproducibility," J Biomed Opt. 13, 054015, 2008, oxygen measurements vary up to 12% with retinal pigmentation, indicating critical influences of SCs from retinal layers like retinal pigment epithelium. Such biases are highly significant and are suggestive of the biases found in the alternative methods tested in Table 2 and Fig. 3 in our manuscript. Garg *et al.* concluded that "Although measurement of peripheral oxygen saturation has become a standard clinical measurement through the development of pulse oximetry, developing a noninvasive technique to measure retinal oxygen saturation has proven challenging, and retinal oximetry technology currently remains inadequate for reliable clinical use."

We added new discussions regarding published 2D oximetry to paragraph 4 of the discussion section in the revised manuscript to clarify the distinction between removing SCs and a calibration procedure.

3. *The section describing the algorithm provides welcome detail, but the absence of the overall context and the bigger picture makes this unengaging and would be more suited as an appendix. Understanding would be enhanced by a top-level discussion of the principles. For the results described in Fig 3A, it seems that conclusions have been reached on a very small data set. There is immense variability in optical clutter and associated fitting both within a single eye, between nominally similar healthy eyes and between eyes that vary with health, ethnicity and age. It is important to account for such variabilities in forming reliable empirical conclusions and this is not adequately addressed here, although the sample size may be reduced with the benefit of a rigorous description of the physics of light propagation in the retina.*

Reply: We thank the reviewer for the suggestion to improve the manuscript. Following the reviewer's suggestion, we revised the first paragraph of the section "Principle of ADS-vis-OCT" to provide a "big picture" summary of the algorithm. The newly added summary will provide readers with a general understanding of the algorithm in the context of the study.

Our dataset includes 125 unique vessels. While Fig. 3 highlights 4 vessels, the conclusion that ADS-vis-OCT eliminates spectral contaminants with normalization at the top of the blood is based on a much larger dataset of 125 unique vessels from 18 patients of different origins and ages. As stated in the paper, the average R^2 for ADS-vis-OCT was 0.96, 0.93, and 0.95 across 21 bands for major arteries, minor arteries, and veins, respectively. If SC were not removed, the fits would fail (Figs. 3B-3I and Table 1). Table 1 summarizes the results from all of those 125 unique vessels, further highlighting the emphasis of the conclusion in Fig. 3. As our recent paper describes the physics of light propagating in blood in much more detail (Fang *et al.*, "Multiple forward scattering reduces the measured scattering coefficient of whole blood in visible-light optical coherence tomography," Biomed. Opt. Express 13, 4510-4527, 2022), we did not add a comprehensive discussion about light propagation but focused on the adaptive method without distracting readers from the key emphasis of the manuscript. The light propagations simulated by Fang *et al.* includes single scattering from blood, multiple forward scattering from blood, the influence of the cell-free zone, and the influence of the vessel wall on light propagation. This work was recently published and cited in our manuscript. We have added statements regarding light propagation in the retina to the third paragraph of the discussion section.

The "immense variability in optical clutter" mentioned underlies one of the key reasons why adaptive normalization by the top blood (mentioned in Fig. 3a) results in better fits than normalization by the NFL or anterior wall. As the optical properties of the retinal nerve fiber layer and the anterior wall may vary between different individuals of varying ethnicity and health, as well as imaging conditions, the relative contribution of these tissues to the vis-OCT signal is highly variable and unpredictably adds spectral contaminants. By adaptively normalizing vis-OCT measurements at the top blood, we can isolate the specific blood signal to the selected blood vessel under current imaging conditions from the SCs associated with outside of the blood retinal layers and rely solely on the optical properties of blood, which are widely validated in literature (Bosschaart *et al.*, "A literature review and novel theoretical approach on the optical properties of whole blood," Lasers Med Sci, 453-79, 2014.) As our manuscript focuses on reporting the methodology of adaptive oximetry rather than a large-scale clinical study, we did not overly emphasize the influence of age, gender, ethnicity, and disease on our measurements which will be addressed in our ongoing studies. However, note that our methodology of adaptive rejections of the SCs automatically reduces variability between different ages, genders, ethnicity, and diseases.

4. *The ability to conduct distinct oximetries in vessels 3 and 4 in Fig 4 is interesting, but the oximetry of 85.8% seems to be much lower than systemic oxygenation seems improbable and merits more rigorous. The hypothesis that lower SO₂ is due to diffusion is not substantiated and could also be due to short comings in measurement accuracy, which is an important claim of the paper. Evidence in support of the proposed hypothesis could be an oxygenation gradient along the length of the vessel – which would be expected from diffusion, since the arterial blood at the exit of the ONH should be close to saturated. Other researchers (eg Einar Steffanson's group) have hypothesised oxygen counter currents along the length of arteries due to oxygen diffusion to veins. Laminar flow also results in variations in oximetry across the widths of some venules (due to different venules at branches draining different parts of the retina with dissimilar oxygen consumptions) - this is not evident in Fig 4A, but would lend support to the robustness of the technique.*

Reply: We thank the reviewer for the constructive comments and agree that it is premature to hypothesize the oxygen-transport nature of the difference in sO₂ levels between vessels 3 and 4. Indeed, a more rigorous investigation is required to support our claim that the lower sO₂ level is due to diffusion, which is out of the scope of this paper. However, we believe showing sO₂ measurements in vessels 3 & 4 is of interest to readers since they depict a unique benefit of vis-OCT's depth resolution over a 2D fundus camera or SLO-based retinal oximetry. This is noted in the "Retinal oximetry around the optic disk" section, which states "We measured sO₂ values from both vessels, demonstrating the unique depth-resolved sO₂ imaging capability permitted by vis-OCT." Whereas non-depth-resolved or low-depth resolved technologies would be unable to separate sO₂ measurements from these two vessels, vis-OCT allows isolation of the signal from each vessel and delineate oximetry measurements. We believe our measurement describes the physical reality of oxygenation differences between these two vessels as our technique has been validated carefully in phantoms, the fits were robust and had high R², and the deeper, larger retinal artery had a high oxygen saturation of 98% consistent with pulse oximetry. We nevertheless agree that using vis-OCT to sense oxygen diffusion is of interest in the field, and will make it the subject of future investigations.

5. What was the diversity of the subjects (age, ethnicity, etc.)?

Reply: We included results from 18 healthy subjects (nine males and nine females) with ages ranging from 21 to 62 years old. The median age is 33.5 years old (average 38.1 years old). Among the 18 subjects, 16 were Caucasian, and 2 were Asian. We added this information to the "vessel selection" section of the manuscript.

6. The paper describes some inconsistencies between measurements of blood optical properties. These differ for polarised and unpolarised light – how is polarization incorporated into the discussions? Most Monte Carlo modelling does not include polarisation and in the ballistic propagation regime relevant to OCT there are significant differences between unpolarised and polarised light propagation. What is the justifications for attributing differences in SSF to "spectral contaminants" rather than the other imperfections in the physical model (see also comments below on RBC alignment). How does the polarisation state within the ex vivo measurements compare to polarisation for OCT in the eye (which is generally less well controlled due to birefringence effects). It is striking that 100% OS values all have SD of 0% - is this because these measurements were artificially limited to 100% If so, I think this should be avoided – it is not uncommon for noise to result in oximetries >100%. Indeed the oxygen dissociation curve prevents OS achieving 100% (>99.95%) so 100% is also non physical, just as is >100%.

Reply: We agree with the reviewer that polarization is an important factor to consider in quantitative backscattering measurements like vis-OCT. To maximize the efficiency of interference of all wavelengths, we placed a polarization controller in the reference arm. We adjusted the polarizations to maximize the image signal before the acquisition by matching reference arm polarization with the polarization of the light returning from the retina. Next, since our approach is depth-resolved we normalize in the blood maximum (BM) region (see steps 3-4 in Fig. 2). This step effectively accounts for polarization-dependent backscattering in the tissue above the blood vessel. Polarization-dependent backscattering in the tissue below the blood vessel does not influence the signal acquired from the blood due to time-of-flight gating in vis-OCT. For non-

depth-resolved or low-depth resolved technologies such as SLO and fundus photography, differences in the polarization state of the out-of-blood tissues influence the relative contribution of scattering to measured blood absorption, as mentioned in *Smith et al., Appl Optics 39, 1183-1193, 2000*.

Additionally, the polarization-dependent scattering from RBCs in our study effectively averaged due to two reasons. First, the light source for our system is a supercontinuum laser with randomly polarized output (rotation of the polarizer at the output of the laser does not lead to intensity variation at the output of the polarizer). This means that during an A-scan exposure time of 38 μs of the line-scan camera in the spectrometer, multiple random polarization states illuminating RBC are effectively averaged. Second, in our recent publication (Fang et al., "Multiple forward scattering reduces the measured scattering coefficient of whole blood in visible-light optical coherence tomography," *Biomed. Opt. Express* 13, 4510-4527, 2022,) we compared the Monte Carlo simulation of vis-OCT retinal oximetry with experimental results, showing that a majority of photons detected from the blood vessel are a multiple scattered photons. Multiple scattering will effectively randomize the light polarization along the depth of the vessel as each scattering event alters polarization. Therefore, by averaging A-lines at multiple positions across several B-scans, we averaged out the influences of polarization on blood attenuation in the vessel as well. We prove this by showing strong agreement between the measured optical properties of simulated blood, *ex vivo* blood, and *in vivo* blood in the same paper. Therefore, any influence of polarization-dependent light scattering in the blood and surrounding tissue as well as polarization alterations in the optical fiber of the system interferometer and the birefringent eye tissue, are effectively average or removed in our study.

Regarding occurrences of $\text{sO}_2 = 100\%$, we note that it is the nature of the non-negative least-squares regression algorithm to bound sO_2 between 0% and 100% (Esposito et al., "Partial least squares algorithms and methods," *WIREs Computational Statistics* 5, 1-19, 2013,) not an artificial threshold limit. The "Oximetry Fitting Model" section has been revised to clarify this. Although $\text{sO}_2 = 100\%$ is technically non-physical compared with the physical limit of 99.95%, for example, we nor any other technology has the precision to differentiate between the two values and such difference is therefore negligible. Furthermore, we recognized potential statistical influences of the upper bound $\text{sO}_2 = 100\%$ in the manuscript. We actively increased the strictness of our quality control when $\text{sO}_2 = 100\%$ to minimize the effect of an $\text{SD} = 0\%$ (see Supplementary information: Adaptive Filtering, Stage 2). Finally, we compared sO_2 values in major arteries to pulse oximeter readings (see Supplementary Information: Table S2), where higher measured sO_2 (e.g. $> 98\%$) typically coincided with such high values from the pulse oximeter.

7. Back scatter from RBCs is also effected by laminar flow causing the alignment of RBCs parallel to the vessel walls causing a characteristic figure-eight B-scan reflectivity pattern within the blood vessels – particularly arteries. How is this accounted for within the model? The paper emphasises the importance of measuring total oxygen delivery to the retina by oximetry of the vessels close to the optic nerve, but omits to mention the fact that total delivery requires also a measurement, or inference, of flow and also that a substantial, but variable (auto-regulated) oxygen supply to the retina is provided by the choroid.

Reply: For the size of the vessels analyzed in this work (diameter $> 35 \mu\text{m}$), multiple scattering dominates the detected signal (Fang et al., "Multiple forward scattering reduces the measured scattering coefficient of whole blood in visible-light optical coherence tomography," *Biomed. Opt.*

Express 13, 4510-4527, 2022.) Therefore, light scatters among red blood cells of different orientations in the vessel. Acquiring A-lines over extended integration times (e.g. 38 μ s in this work) and multiple B-scans averages multiple-scattering events from different red blood cell orientations and reduces systemic optical measurement bias. Still, we agree with the reviewer that the potential influence of such well-organized RBC orientations in the laminar flow on vis-OCT sO₂ measurements needs to be considered. In this work, we validated the sO₂ values in the large arteries with a pulse oximeter reading. If these flow patterns invalidated our model, significant discrepancies between pulse oximeter readings and our measurements in arteries would be seen, which is not the case. Organized patterns can potentially bias measurements in smaller vessels, but this is beyond the scope of our work and a subject of further investigation. To account for factors biasing the amount of backscattering, we adapted the fitted scattering coefficient of blood (scattering scaling factor, or SSF) in our model. By populating a distribution of measurements with different SSFs (step 10 in Fig. 2), we statistically estimate the mostly likely sO₂ value while minimizing assumptions of factors like flow and blood cell orientation.

We agree that retinal metabolism depends both on oxygenation and blood flow; however, this subject is beyond the scope of this work and is a subject of future investigation. This study focuses on achieving robust oximetry measurements, making future oxygen metabolism feasible when accurate and robust retinal blood flow measuring technology is developed and validated.

8. *How were the measurements in vessel segments aggregated into a single measurement – was it a single mean or was there a process to remove outliers?*

Reply: We input each B-scan from a vessel segment into a quality control algorithm (Step 5, Fig. 2). B-scans not passing the quality control metrics were removed from the analysis. High-quality B-scans were then averaged together to compute the final measurement. This process is described in the original manuscript section "Principle of ADS-vis-OCT" and in the supplementary information section "Adaptive Filtering (Stage 1)".

The quality control metric is detailed as: "We averaged NL-SDA-lines along a 32 μ m depth region beyond z_n for each respective B-scan to obtain a 1D STFT spectrum. We calculated sO₂ and spectral fit R² from NL-SDA-lines in each B-scan by least-squares fit (see **Methods – Oximetry fitting model**). Then, we applied a threshold of sO₂ > 15% and R² > 0.40 and removed B-scans that did not pass. For the smallest analyzed vessels (diameter < 60 μ m), we did not perform this step, due to increased noises in individual B-scans."

Minor issues

9. *" $\sqrt{I_{sample}(\lambda)}$ and $\sqrt{I_{Ref}(\lambda)}$ are the power spectra of the light collected from the sample and reference arms, respectively." – doesn't the square root make these amplitude spectra?*

Reply: We agree with the reviewer and revised this in the manuscript to:

" $I_{sample}(\lambda)$ and $I_{Ref}(\lambda)$ are the power spectra of the light collected from the sample and reference arms, respectively."

10. *"The term $-\mu_{t_{blood}}(\lambda) \left[(z_0 - z_d) + \frac{\Delta z}{2} \right]$ is a constant and represents uniform, oxygen dependent attenuation along the vessel depth. Blood cell packing, orientation, flow, and oxygen diffusion may*

add variability to this assumption." It could be highlighted that due to the aforementioned laminar flow, oxygenation may vary substantially within venules.

Reply: We agree with the reviewer that this term may vary with the laminar flow in venules. Therefore, mentioned laminar flow in the venules. The sentence has been adjusted to "The term $-\mu_{\text{tblood}}(\lambda) \left[(z_0 - z_d) + \frac{\Delta z}{2} \right]$ is a constant and represents uniform, oxygen-dependent attenuation along the vessel depth. Blood cell packing, orientation, flow, and specifically laminar flow in venules, and oxygen diffusion may add variability to this assumption."

11. *"consistent with our previous findings ()" – missing reference*

Reply: Citation "Raymond Fang, Ian Rubinoff, and Hao F. Zhang, "Multiple forward scattering reduces the measured scattering coefficient of whole blood in visible-light optical coherence tomography," Biomed. Opt. Express 13, 4510-4527 (2022)" has been added.

REVIEWERS' COMMENTS:

Reviewer #1 (Remarks to the Author):

I thank the authors for their thorough revision.

Reviewer #4 (Remarks to the Author):

See attachment.

Review of “*Adaptive Spectroscopic Visible Light OCT for Clinical Retinal Oximetry*” by Rubinoff et al.

General Comment

This paper examines the unwanted signals present in visible light OCT that affect the measurement accuracy and repeatability of blood oxygenation. The paper is well written, clear, concise and is an important step towards clinical translation and suitable for publication in Nature Communications Medicine.

There is increasing interest in coupling the high structural resolution of OCT with function measurements in the retina for example as in oxygen saturation explored in this manuscript to optoretinograms looking at cone responses in response to a light stimulus. While the subject population is small and the big question of repeatability across different populations is outside the scope of this paper, the discussion of the factors affecting the measurement of retinal blood oxygenation are well made and the paper is very timely.

As requested I have focused on the responses to reviewer 3.

Specifically:

Reviewer 3, Comment 1.

I agree with the comment raised by Reviewer 3 that the correlation between oxygenation and disease detection sensitivity is potentially overstated (at least at the current time). The recent work of Don Miller for example has demonstrated the ability to image individual retinal ganglion cells and monitoring this cell type over time may be one of the earliest indications of glaucomatous damage, possibly preceding vascular changes. The last sentence of the first paragraph of the introduction should therefore be modified to say that oxygen saturation has the potential to be a sensitive biomarker for several retina diseases....

Reviewer 3, Comment 2.

The authors have provided a thorough response to the points raised by the Reviewer on the accuracy of vis-OCT compared to fundus camera or SLO oximetry approaches.

The term 'spectral contaminants' would appear to be appropriate here.

Can the authors comment on potential future steps to improve accuracy and repeatability of vis-OCT for example adaptive optics for precise control of beam shape, use of a lower NA and hence a less aberrated imaging beam, image stabilization methods. Maybe these are being implemented by the team? While they may be impractical in a clinical system they may have value to further refining the SC reduction techniques.

Reviewer 3, Comment 3, comment 5.

The diversity / refractive errors etc. of the subjects is important and should be provided in a separate paragraph instead of being included in 'Vessel Selection' on page 16.

Reviewer 3, Comment 4.

This has been thoroughly addressed by the authors.

Reviewer 3, Comment 6.

As the Reviewer mentions polarization is an important factor to consider when making quantitative measurement in OCT. An abbreviated version of the author response to this comment would a good addition to the Supplementary Information document.

Reviewer 3, Comments 7 & 8.

These are thoroughly addressed by the authors.

Minor Comments

Top of page 3 / Figure 1. 'Number 1' is used in the figure whereas 'group 1' is used in the text. I believe this refer to the same photon path but this should be revised for clarity.

Reviewer #4

Review of “Adaptive Spectroscopic Visible Light OCT for Clinical Retinal Oximetry” by Rubinoff et al.

General Comment

This paper examines the unwanted signals present in visible light OCT that affect the measurement accuracy and repeatability of blood oxygenation. The paper is well written, clear, concise and is an important step towards clinical translation and suitable for publication in Nature Communications Medicine.

There is increasing interest in coupling the high structural resolution of OCT with function measurements in the retina for example as in oxygen saturation explored in this manuscript to opto retinograms looking at cone responses in response to a light stimulus. While the subject population is small and the big question of repeatability across different populations is outside the scope of this paper, the discussion of the factors affecting the measurement of retinal blood oxygenation are well made and the paper is very timely. As requested I have focused on the responses to reviewer 3.

Reply: We thank this reviewer for the appreciation of our work.

Specifically:

Reviewer 3, Comment 1.

I agree with the comment raised by Reviewer 3 that the correlation between oxygenation and disease detection sensitivity is potentially overstated (at least at the current time). The recent work of Don Miller for example has demonstrated the ability to image individual retinal ganglion cells and monitoring this cell type over time may be one of the earliest indications of glaucomatous damage, possibly preceding vascular changes. The last sentence of the first paragraph of the introduction should therefore be modified to say that oxygen saturation has the potential to be a sensitive biomarker for several retina diseases....

Reply: We thank the reviewer for the input. We have revised the last paragraph of the introduction accordingly.

Reviewer 3, Comment 2.

The authors have provided a thorough response to the points raised by the Reviewer on the accuracy of vis-OCT compared to fundus camera or SLO oximetry approaches. The term ‘spectral contaminants’ would appear to be appropriate here.

Can the authors comment on potential future steps to improve accuracy and repeatability of vis-OCT for example adaptive optics for precise control of beam shape, use of a lower NA and hence a less aberrated imaging beam, image stabilization methods. Maybe these are being implemented by the team? While they may be impractical in a clinical system they may have value to further refining the SC reduction techniques.

Reply: A paragraph on potential future steps, including varying NA, was added as the second to last paragraph of the discussion.

Reviewer 3, Comment 3, comment 5.

The diversity / refractive errors etc. of the subjects is important and should be provided in a separate paragraph instead of being included in 'Vessel Selection' on page 16

Reply: An additional section entitled 'Subject Demographics' was added, which includes subjects' age, gender, race, and how recruitment was done. Unfortunately, we do not have data on refractive errors.

Reviewer 3, Comment 4.

This has been thoroughly addressed by the authors.

Reviewer 3, Comment 6.

As the Reviewer mentions polarization is an important factor to consider when making quantitative measurement in OCT. An abbreviated version of the author response to this comment would a good addition to the Supplementary Information document.

Reply: We have added a section entitled "Influence of Polarization in vis-OCT Retinal Oximetry" to the supplemental information.

Reviewer 3, Comments 7 & 8.

These are thoroughly addressed by the authors.

Minor Comments

Top of page 3 / Figure 1. 'Number 1' is used in the figure whereas 'group 1' is used in the text. I believe this refer to the same photon path but this should be revised for clarity.

Reply: Figure 1 has been adjusted to say group 1 for consistency purposes.